# Scale-Wise VAR is Secretly Discrete Diffusion

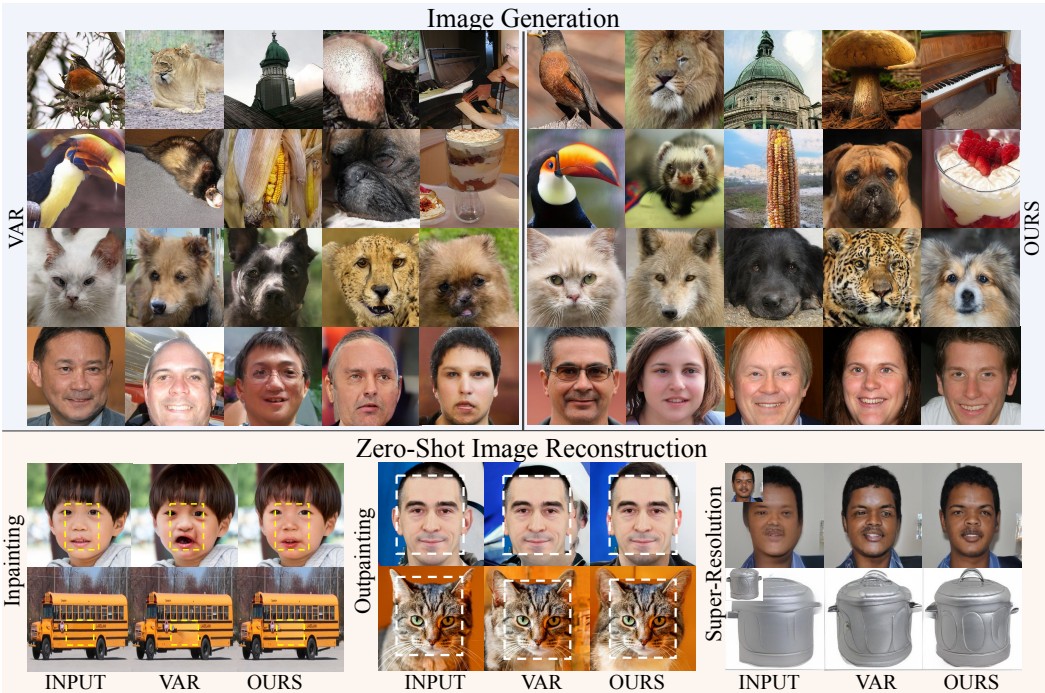

Figure 1: **Figure illustrating the different applications of SRDD method:** SRDD exhibit better sampling fidelity and zero shot performance compared to the VAR.

## ABSTRACT

Autoregressive (AR) transformers have emerged as a powerful paradigm for visual generation, largely due to their scalability, computational efficiency and unified architecture with language and vision. Among them, next scale prediction Visual Autoregressive Generation (VAR) has recently demonstrated remarkable performance, even surpassing diffusion-based models. In this work, we revisit VAR and uncover a theoretical insight: when equipped with a Markovian attention mask, VAR is mathematically equivalent to a discrete diffusion. We term this reinterpretation as Scalable Visual Refinement with Discrete Diffusion (SRDD), establishing a principled bridge between AR transformers and diffusion models. Leveraging this new perspective, we show how one can directly import the advantages of diffusion—such as iterative refinement and reduce architectural inefficiencies into VAR, yielding faster convergence, lower inference cost, and improved zero-shot reconstruction. Across multiple datasets, we show that the diffusion-based perspective of VAR leads to consistent gains in efficiency and generation. To facilitate further research, we will make the code and models public.

## 1 INTRODUCTION

Autoregressive models Bengio et al. (2003); Papamakarios et al. (2017) are among the most efficient and scalable approaches for generative modeling van den Oord et al. (2016b); Brown et al. (2020); van den Oord et al. (2016a). Recent work Austin et al. (2021) shows that autoregressive training

can be viewed as a discrete diffusion variant, where tokens are masked in a fixed order rather than randomly as in diffusion. However, using this formulation for visual generation introduces two key limitations: (i) the autoregressive paradigm introduces an inductive bias, where pixels or regions generated initially are not informed of the distribution or semantics of the generated image. (ii) the model receives no explicit signal about the degree of degradation, forcing it to learn this internally (one could imagine this as the initial tokens having more degradation and the final ones having less). As a result, despite their efficiency, AR models underperform when directly combined with diffusion-style training strategies.

Although effective for text generation, these models have not been successful for image generation. Diffusion models Ho et al. (2020); Song et al. (2020; 2023) have portrayed the capability to generate high quality images by iteratively denoising pure noise to a point in the data distribution through a large number of steps. Although effective for generating high-quality images, these models are notoriously slow Peebles & Xie (2023); Chang et al. (2022); Saharia et al. (2022) and require extensive design choices Rombach et al. (2022); Peebles & Xie (2023); Salimans (2016); Song et al. (2023); Lu et al. (2023; 2022); Song et al. (2021) for fast inference. Moreover, increasing the model size for the diffusion model leads to heavy inference

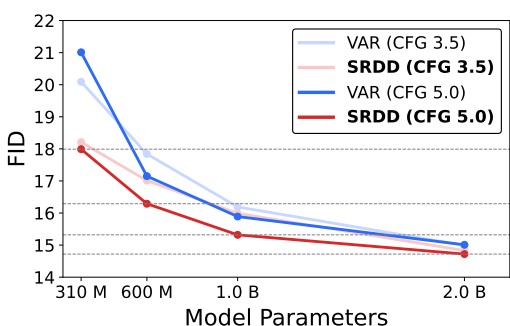

Figure 2: **Scaling behaviour of SRDD and VAR:** SRDD exhibits similar scaling behavior with parameter size as observed in VAR

computational requirements to achieve good quality results. Tackling the fundamental limitations of a diffusion model requires a model that can perform fast generation while exhibiting scalability with compute and parameter size. Recently, Visual Autoregressive Generation Tian et al. (2024) (VAR) introduced a new paradigm of models based on next scale prediction using transformers. These models, rather than predicting the next token as in GPT architectures Chen et al. (2020); Sun et al. (2024); Ramesh et al. (2021), autoregressively predict the next scale corresponding to a higher-resolution image. Moreover, VAR has also shown that increasing the parameters of the model drastically improves the generation quality in terms of FID scores.

In this work, we delve deep into the inner workings of VAR and discrete diffusion models. We observe similar findings of existing work Voronov et al. (2024), suggesting that the current version of VAR has design inefficiencies and the overall model can be improved further by predicting the next scale in a Markovian fashion, conditioned on the immediate previous scale rather than all previous scales. Our analysis of the training dynamics and the loss functions of the model reveals that the *Markovian variant of VAR is an efficient formulation of a discrete diffusion model*. Motivated by this, we present Scalable Visual Refinement with Discrete Diffusion (SRDD), a theoretical perspective that interprets the Markovian variant of VAR, together with probabilistic sampling techniques, through the lens of discrete diffusion. To the best of our knowledge, we are the first work to connect a variant of VAR to a discrete diffusion. As shown in (Figure 2), SRDD inherits VAR's strong scaling behaviour, achieving improved performance with increasing model size. The discrete diffusion perspective brings in an added benefit, such as utilizing all relevant literature holding for discrete diffusion models in VAR formulation. This in turn, drastically improves the generation quality of VAR without the need for explicit handcrafted design choices, but instead uses structured choices deep-rooted in theory.

We experiment with three different properties tied to probabilistic sampling with diffusion properties, such as (1) classifier-free guidance Ho (2022); Schiff (2024) (2) token resampling Wang et al. (2025), and (3) distillation Salimans & Ho (2022); Meng et al. (2023), and show that SRDD in turn works better when combined with these strategies. Moreover, like diffusion models, we also explore zero-shot generation performance like super-resolution, inpainting, and outpainting and obtain better results than the original VAR architecture. We present these results in Figure 1. With this work, we reveal a new perspective on VAR by formally connecting it to discrete diffusion with a theoretical lens that explains its behaviour and informs principled design choices, and direct the attention of the community to how the quality, efficiency and explainability of visual generation can be further improved. Thereby, we open up possibilities in visual generation research. This explainability may be further used for design choices while scaling up LLMs for joint visual-language generation as well.

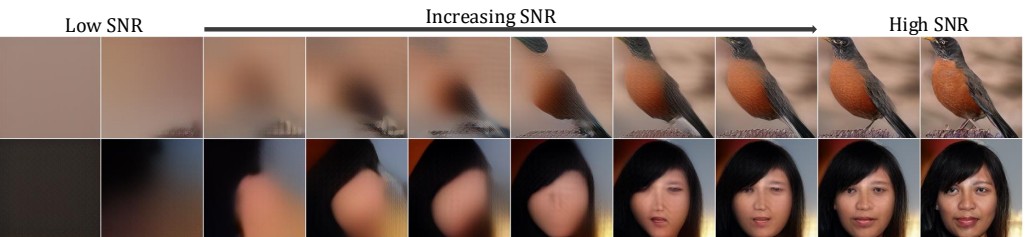

Figure 3: **Scale-wise generation of VAR:** The SNR increases through the generation process, similar to the diffusion process.

## 2 BACKGROUND

In this section, we describe in brief detail the working of visual autoregressive generation and discrete diffusion models.

**Visual Autoregressive Generation:** Tian et al. (2024) brought about a new paradigm for visual generative modeling, where the model is trained for next-scale prediction. Unlike earlier autoregressive models that generate discrete tokens at a single resolution sequentially, in VAR, all tokens at one resolution are generated jointly, and then progressively refined to move from the lowest to the highest resolution. To generate an image with resolution $H \times W$, the generation process happens progressively through sub-resolutions $x_i = h_i \times w_i$. At each step, the model conditions on all previously generated resolutions, effectively modeling $p(x_1, x_2, ..., x_i) = \prod_{i=1}^{N} p_\theta(x_i | x_{i-1}, x_{i-2}, .., x_1)$, where $x_i$ denotes discrete tokens corresponding to different resolutions obtained through the multi-scale VQVAE van den Oord et al. (2016b). These tokens by themselves may not form any meaningful image, but the summation of residual over different resolutions reconstructs the whole image. An autoregressive transformer is trained to learn the corresponding distribution. The effective training loss for VAR can be written as

$$\mathcal{L} = -E_{q(x_N)} \left[ \Sigma_{i=1}^{N} \log p_\theta(x_i | x_{i-1}, ..., x_1) \right], \tag{1}$$

where $N$ is the total number of resolutions in the generation and $q()$ is the training data distribution.

**Discrete diffusion models:** are the discrete counter parts of continuous-time diffusion models. These models were first proposed inSohl-Dickstein et al. (2015), then later extended in Sahoo et al. (2024); Hoogeboom et al. (2022); Luo et al. (2023a). D3PMsAustin et al. (2021) elaborated more on discrete diffusion models and brought in the new perspective of rethinking the transition noise matrices. In a general discrete diffusion model, the transition between adjacent states is modelled as a categorical distribution, where the current state is transformed through a transition matrix. We formally define this by $q(x_t | x_{t-1}) = Cat(x_t | p = x_{t-1} Q_t)$, where $Q_t$ is the transition matrix from a state $x_{t-1}$ to a state $x_t$ and $q(x_t | x_0) = Cat(x_t | p = x_0 \overline{Q_t})$, where $\overline{Q_t} = Q_1 Q_2 \cdots Q_t$. The choice of the transition matrix decides the nature of degradation existing in the diffusion process and is designed by $[Q_t]_{ij} = q(x_t = j | x_{t-1} = i)$. Like in a continuous time diffusion model, a parameterized model $p_\theta(x_t, t)$ learns the reverse distribution, removing degradation from an input signal $x_t$, given the amount of degradation. Discrete diffusion models are trained with cross-entropy loss predicting the categorical distribution at each timestep $t$, formally defined as,

$$\mathcal{L} = -E_{q(x_0)} \left[ \Sigma_{t=1}^{T} E_{q(x_t | x_0)} \left[ \log p_\theta(x_0 | x_t) \right] \right]. \tag{2}$$

Alternatively, though the Markovian formulation, diffusion models may also be trained to reconstruct $x_{t-1}$ given $x_t$ directly using the parameterized model. The corresponding loss function is written as

$$\mathcal{L} = -E_{q(x_0)} \left[ \Sigma_{t=1}^{T} E_{q(x_t | x_0)} \left[ \log p_\theta(x_{t-1} | x_t) \right] \right], \tag{3}$$

where $p_\theta$ is a diffusion model that iteratively restores a sample from the degraded distribution to one in the tokens of real distribution.

**Concurrent works:** Recent works, Kumbong et al. (2025); Voronov et al. (2024), observe that VAR assumes all preceding scales are equally important for generating the next scale, even though the current resolution already encodes prior-scale information, making such conditioning redundant and architecturally inefficient as shown in Figure 3. While prior work recognizes these shortcomings, it lacks a theoretical explanation of why Markovian variants perform better. In this paper, we bridge this gap by showing that the Markovian formulation of VAR naturally aligns with the discrete diffusion perspective, thereby offering a principled explanation for the observed performance gains.

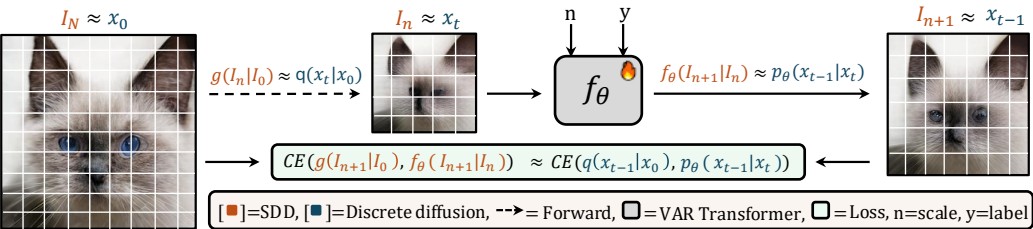

Figure 4: **Figure illustrates the connection between the Markovian variant of VAR (SDD) and discrete diffusion.:** The SDD forward process $g(I_n \mid I_0) = M(n)I_0$ mirrors the diffusion transition $q(x_t \mid x_0)$, where the ground truth $I_0$ is deterministically degraded by the transition matrix $M(n)$. Further, the learnable transformer $f_\theta(I_n, n, y)$ predicts the coarser-to-finer transition $I_{n+1}$, analogous to the reverse diffusion step $p_\theta(x_{t-1} \mid x_t)$. Importantly, the training objective in both cases reduces to a cross-entropy loss between the forward posterior and model prediction, making the loss formulation of SDD equivalent to the diffusion ELBO in the limiting case of a deterministic transition.

## 3 METHOD

In this section, we connect the working of Markovian variant VAR (referred as SDD) to that of a discrete diffusion model as shown in Fig. 4.

### 3.1 AUTOREGRESSIVE MODELS AS DISCRETE DIFFUSION MODELS.

Following Austin et al. Austin et al. (2021), an autoregressive process can be interpreted as a special case of a discrete diffusion model. Consider a sequence of length $N = T$ and a deterministic forward process that progressively masks tokens one by one $q([x_t]_i \mid x_0) = [x_0]_i$ if $i < T - t$ else [MASK]. This implies that $q(x_{t-1} \mid x_t, x_0)$ is a delta distribution over the sequence with one fewer mask: $q([x_{t-1}]_i \mid x_t, x_0) = \delta_{[x_t]_i}$ if $i \neq N - t$ else $\delta_{[x_t]_0}$. Although this procedure does not act independently on each token, it can be recast as a diffusion process defined over the product space $[0, N] \times \mathcal{V}$, where $\mathcal{V}$ is the vocabulary and $\mathbf{Q}$ is an $N \times |\mathcal{V}| \times N \times |\mathcal{V}|$ sparse transition matrix. All tokens except the one at position $i = T - t$ have deterministic posteriors, so the KL divergence

$$D_{\mathrm{KL}}\big(q([x_{t-1}]_j \mid x_t, x_0) \,\|\, p_\theta([x_{t-1}]_j \mid x_t)\big) = 0, \quad \text{for } j \neq i \tag{4}$$

vanishes for $j \neq i$. The only non-trivial divergence occurs at position $i$, yielding

$$D_{\mathrm{KL}}\big(q([x_{t-1}]_i \mid x_t, x_0) \,\|\, p_\theta([x_{t-1}]_i \mid x_t)\big) = -\log p_\theta([x_0]_i \mid x_t), \tag{5}$$

which exactly corresponds to the standard cross-entropy loss used in autoregressive training.

### 3.2 RETHINKING VAR VARIANTS THROUGH THE DISCRETE DIFFUSION LENS

To illustrate how VAR is a variant of discrete diffusion models, we link VAR towards the key characteristics of discrete diffusion models (1) A model parameterized with the amount of degradation to remove (2) A categorical distribution matching loss function (3) A progressively increasing SNR during the generation process

**(1) A amount of degradation parameterized into the model input:** VAR inherently is an iterative refinement model trained to reconstruct tokens of different levels of intensities. Just as in diffusion models where the timestep of diffusion is conditioned to the model, we found out that in the original implementation for VAR, the current resolution(scale) to be restored is parameterized, embedded and informed through the model through a concatenation operation along with the class embedding.

**(2) Loss function for training VAR:** Another notable design choice of VAR is the use cross-entropy loss for predicting discrete tokens as defined in 1. Taking a closer look at the loss for discrete diffusion where categorical distribution matching happens through cross entropy loss, Equation (3),

$$\mathcal{L} = -E_{q(x_0)}\left[\Sigma_{t=1}^T E_{q(x_t \mid x_0)}\left[\log p_\theta(x_{t-1} \mid x_t)\right]\right]. \tag{6}$$

In the limiting case where there is only one possible transition between the states $x_t \to x_{t-1}$. And the final stationary state $\mathbf{x_T}$ is predefined to a fixed $< SOS >$ token, the effective loss function becomes,

$$\mathcal{L} = -E_{q(x_0)}\left[\Sigma_{i=1}^T \log p_\theta(x_{i-1} \mid x_i)\right]. \tag{7}$$

Taking a closer look at 1. We find that in the limiting case of a deterministic transition matrix, this is the exact same loss function(within the factor of a scaling constant) used to train VAR, but rather conditioned on the previous scale alone.

**(3) Progressively increasing SNR**: We reformulate VAR as a model that recursively reconstructs images of higher scales conditioned on low scales. The low resolution tokens $I_n \in R^{n \times n}$ at a scale $n$, are obtained through downsampling from tokens of resolution $I_N \in R^{N \times N}$ through $I_n = M(n).I_0$, $M(n) \in R^{n^2 \times N^2}$ is a matrix that performs a non-linear deterministic downsampling operation dependent on $n$ and N is the maximum scale. At each scale $n$, the model $f_\theta$ predicts the residual relative to the upsampled previous scale,

$$\mathbf{f_\theta(I_{n-1}, n)} : (I_{n-1})_{\uparrow(n)} \to I_n - (I_{n-1})_{\uparrow(n)}, \tag{8}$$

where $(I_{n-1})_{\uparrow(n)}$ denotes a upsampling operation that upscales $I_{n-1}$ to the size of $I_n$. The exact transformation is provided in the supplementary material.

As the scale index $n$ increases, the signal-to-noise ratio (SNR) of $I_n$ also increases, with smaller $n$ corresponding to coarser, noisier resolutions. Thus, the progressive downsampling of the original image $I_N$ into multiple resolutions can be interpreted as a diffusion process as prescribed in D3PMs Austin et al. (2021) with deterministic transition matrix $\mathbf{Q}$ as $\mathbf{M(n)}$. This behaviour is illustrated in Figure 3, showing how SNR improves through successive stages of the generation process.

The corresponding transformation for a diffusion model, for the transition from a state $x_t \to x_{t-1}$, brings in an effective transformation,

$$\sqrt{\alpha_t}x_0 + \sqrt{(1-\alpha_t)}\epsilon_1 \to \sqrt{\alpha_{t-1}}x_0 + \sqrt{(1-\alpha_{t-1})}\epsilon_2; \epsilon_1, \epsilon_2 \sim \mathcal{N}(0, I) \tag{9}$$

The extra information on a signal level brought by the model can be described as

$$\mathbf{p_\theta(x_t, t)} : \sqrt{\alpha_t}x_0 + \sqrt{(1-\alpha_t)}\epsilon_1 \to (\sqrt{\alpha_{t-1}} - \sqrt{\alpha_t})x_0, \tag{10}$$

where $\sqrt{\alpha_{t-1} - \alpha_t} \to 0$ as $t \to 0$. Here $\mathbf{p_\theta}(.)$ is the diffusion model bringing this transformation. Comparing Equation (10) and Equation (8), we see that in both models, a parameterized network learns the residual signal information required at a particular SNR.

A model satisfying all the above three criteria could be broadly categorized as a difffusion modelBansal et al. (2023). However, conventional diffusion-based realization of this approach based on existing literature would ideally require all corresponding latents $x_t$ to be of the same resolution. Although more efficient methods have been proposedTeng et al. (2023); Zheng et al. (2024), these methods still operate at a small number of resolutions, with each resolution having multiple diffusion steps. Here is where the efficiency of VAR comes into play. If VAR can be modified to be dependent on the previous scale alone, an efficient modelling of a discrete diffusion process becomes possible. This could be performed by converting the blockwise causal mask to a Markovian attention mask. The Markovian variant of VAR outperforms VAR over multiple datasets. We argue that this observation is because the Markovian variant of VAR acts like the exact formulation of a discrete diffusion model, resulting in a higher evidence lower bound (ELBO) than the autoregressive formulation. We refer to this model—with fixed start token, fixed transitions, and Markovian attention—as **Scalable Discrete Diffusion (SDD)**, and validate its effectiveness through distribution-matching experiments.

This observation further opens up multiple possibilities (1) An explainability aspect to VAR that connects it to discrete diffusion models, which suggests possibilities for how to better boost performance. (2) All the works and numerous research papers for enhancing discrete diffusion models can now be utilized for VAR variants for enhanced generation process. (3) VAR showed that the utilization of properties like classifier-free guidance(cfg) and scaling model size improved performance, but this was an empirical observation. We inturn explain why these design choices brought in improvements and how we can further enhance the performance of the models.

In the next section, we detail four different variants of design choice that can significantly boost the performance of a Markovian variant of VAR. Many of these are motivated by their counterparts from continuous and discrete diffusion models

### 3.3 How to Improve the Generation Performance and Efficiency

We present four different methods for enhancing the performance of our Markovian version of VAR:

Table 1: **Quantitative results compared to different generative models on the same training setting:** We compare using FID and IS on conditional and unconditional generation tasks. Here, "-" denotes that the model has not converged during the training process.

| Method | Conditional | | | | Unconditional | | | |
|---|---|---|---|---|---|---|---|---|
| | MiniImageNet | | SUN | | FFHQ | | AFHQ | |
| | FID($\downarrow$) | IS($\uparrow$) | FID($\downarrow$) | IS($\uparrow$) | FID($\downarrow$) | IS($\uparrow$) | FID($\downarrow$) | IS($\uparrow$) |
| LDM | 84.13 | 15.79 | 34.62 | 17.69 | 18.91 | 3.95 | 92.53 | 5.09 |
| DiT-L/2 | 57.55 | 31.29 | – | – | 28.44 | 3.51 | – | – |
| VAR | 21.01 | 59.32 | 15.72 | 16.19 | 19.23 | 3.09 | 14.74 | 9.92 |
| **Ours: SRDD** | **16.76** | **63.31** | **13.26** | **17.97** | **17.37** | **4.05** | **13.14** | **10.09** |

**(a) Classifier free guidance:** Classifier-free guidance (cfg) has been widely studied in diffusion models. Ho (2022) provided a probabilistic interpretation, showing that at each sampling step the model generates outputs biased toward the conditional distribution while being pushed away from the unconditional data distribution. This is defined formally as $p(x|c) \sim \frac{p^{w+1}(x|c)}{p^w(x|\phi)}$ where $\phi$ denotes the unconditional distribution. In VAR, cfg was previously tuned in an ad-hoc manner, yielding an empirical "optimal" value but without a consistent trend. In contrast, we show the effect of cfg for SDD and the naive VAR model, as we can observe, making the model Markovian and presenting the discrete diffusion perspective brings in a behaviour pattern for different cfg values and enables to boost performance higher, similar to that observed in diffusion models.

**(b) Token resampling for enhanced generation:** Recent works in discrete diffusion for language generation Nie et al. (2025); Sahoo et al. (2024) propose resampling low-probability tokens at each timestep conditioned on the remaining context. We adopt this strategy in SDD, calling it Masked Resampling (MR) and final models as SRDD: at each resolution in SDD, tokens with prediction probability below 0.01 are resampled multiple times to improve generation quality. This process refines the out-of-distribution tokens at each stage.

**(c) Simple resampling for enhanced generation.** Diffusion models also benefit from increasing the number of sampling steps. Analogously, we enhance SDD by performing multiple sampling steps per scale, effectively increasing the refinement depth.

**(d) Distillation of VAR variants:** Distillation of diffusion models has been extensively studied. Starting with progressive samplingSalimans & Ho (2022), DMDYin et al. (2024), multiple distillation methodsMeng et al. (2022; 2023) have been proposed for more efficient generation. In a similar fashion we explore the effectiveness of progressive distillation in our variant of VAR. Starting with a pretrained SDD model, as in diffusion, we skip certain scales as the distillation proceeds, which inherently increases the SNR gap between consecutive scales. To replicate this in SDD, we drop certain resolutions in VAR and upsample the previous resolution for the discrete latent tokens: $h_i, w_i \rightarrow h_{i+m}, w_{i+m}; m > 1$. We provide further analysis in the experiments section.

## 4  EXPERIMENTS

**Implementation Details.** We use the decoder-only Transformer design of VAR Tian et al. (2024). To enforce the scale-wise Markovian dependency described above, we replace the block-wise causal mask with a Markovian mask that lets tokens at scale $s$ attend to all tokens from scale $s-1$. We reuse the codebook and tokenizer of VAR: a single VQ codebook with vocabulary size $V = 6,000$ shared across all scales. The codebook is frozen during Transformer training. All models are trained with AdamW ($\beta_1 = 0.95$, $\beta_2 = 0.05$, weight decay 0.05) and a learning rate of $10^{-4}$. We employ a batch size of 224 and clip gradients at a norm of 1.0. Training runs for 200 epochs on 4 NVIDIA A6000 GPUs. Apart from the Markovian mask and resampling, every hyper-parameter is kept identical to the VAR configuration to ensure a fair comparison. **In our academic setting, we are limited to a modest GPU budget; consequently, all ablations are conducted on the reduced datasets.**

### 4.1  EXPERIMENT RESULTS

Table 1 shows the comparison of SRDD with three strong generative baselines—LDM Rombach et al. (2022), DiT-L/2 Peebles & Xie (2023), and the VAR Tian et al. (2024) which are Pre-trained with 200 epochs—on four different benchmarks. SRDD approach yields the best FID and IS on

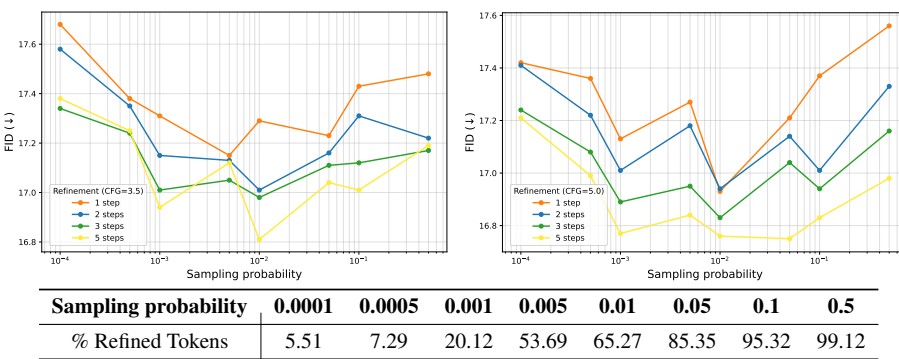

| Sampling probability | 0.0001 | 0.0005 | 0.001 | 0.005 | 0.01 | 0.05 | 0.1 | 0.5 |
|---|---|---|---|---|---|---|---|---|
| % Refined Tokens | 5.51 | 7.29 | 20.12 | 53.69 | 65.27 | 85.35 | 95.32 | 99.12 |

Figure 5: **Ablation study illustrating the effect of MR:** We experiment with different threshold $p_{\text{resample}}$ and the number of refinement steps (Zoom in for better view)

every dataset. Against the VAR, we observe that our method has relative FID drops of **20.2%** on MiniImageNet Dhillon et al. (2019), (21.01→16.76), **9.7%** on FFHQ Karras et al. (2019)(19.23 →17.37). These improvements are accompanied by IS gains of **6.7%** and **31.1%**, respectively. DiT-L/2 and LDM trail far behind—e.g. on MiniImageNet DiT-L/2 obtains an FID of $57.55$ and LDM achieves $84.13$ , more than $3\times$ worse than ours—highlighting the data-efficiency advantage of our scale-wise Markovian design. We use the same number of epochs (200) to train all the models.

Figure 1 visualizes random generations from VAR (left block) and SRDD (right block). Across all three domains—MiniImageNet (top row), FFHQ (middle), and AFHQ (bottom)—our images exhibit noticeably sharper edges, cleaner textures and far fewer structural artifacts: The bird, lion and mice images from VAR suffer from blurred contours and texture collapse, whereas ours preserve fine feather patterns and realistic fur. Faces generated by SRDD contain consistent skin tones and symmetric facial features; VAR often produces mottled skin and asymmetries. Animal portraits (e.g., cat, dog, leopard) demonstrate higher fidelity in ear positioning, eye clarity and background coherence with our approach. Additional examples are provided in the supplementary material.

## 4.2 METHOD-WISE ANALYSIS

**Resampling** We perform token–level re-sampling during inference. At each refinement step, we (i) compute the acceptance probability for every latent token, (ii) re-sample tokens whose probability falls below a threshold $p_{\text{resample}}$, and (iii) feed the updated grid back into the scale-wise Transformer decoder for another pass. We ablate two factors: the threshold $p_{\text{resample}} \in$

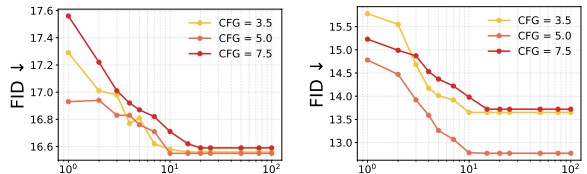

Figure 6: **Effect of refinement steps in MR:** Increasing MR steps leads to convergence.

$\{10^{-4}, \cdots, 10^{-1}\}$ (Fig. 5) and the number of refinement iterations $T \in \{10^0, \cdots, 10^2\}$ as shown in (Fig. 6), under guidance scale, cfg $= 3.5, 5.0, 7.5$.

In Fig. 5 (top), FID decreases monotonically with the number of refinement steps for every threshold and on both cfg values. Across thresholds, the global optimum is reached at $p_{\text{resample}} = 0.01$: after $T = 5$ iterations we obtain an FID of $16.76$ (cfg 5.0) and $16.81$ (cfg3.5). This setting refines $\approx 65\%$ of tokens per pass, striking a balance between coverage (enough tokens are revisited) and context preservation (35% of high-confidence tokens remain to guide the Transformer attention). Lower probabilities ($p_{\text{resample}} < 0.005$) leave too many erroneous tokens untouched, whereas aggressive thresholds ($p_{\text{resample}} \geq 0.05$) remove excessive context, leading to noisy conditioning and a mild FID regression. The trend is consistent across both guidance scales, indicating that the resampling mechanism interacts weakly with classifier-free guidance itself.

Fixing $p_{\text{resample}} = 0.01$, Fig. 6 reveals that when we increase the inference time, most of the quality gains occur in the first 15–25 passes; FID curves flatten afterwards on both MiniImageNet and SUN. This insight suggests that the vast majority of tokens reach the acceptance threshold within 15-25 iterations; subsequent passes bring negligible improvements.

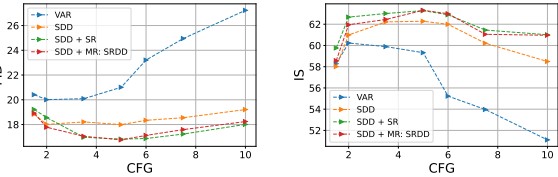

Figure 7: **Qualitative results illustrating impact of different components:** We present the results with each component and their impact.

We also perform simple resampling, inspired by self-refinement in diffusion models, where increasing the number of refinement steps improves quality. Similarly, SRDD benefits from additional self-refinement, as illustrated in Figure 7, which visualizes the contribution of each component to perceptual quality: VAR frequently distorts global geometry (warped goose torso, blurred dog muzzle) and leaves background noise. SDD conditions on the immediate scale, corrects semantics and coarse layout, yet results remain soft and lack high-frequency detail. SDD + SR,: resampling all tokens each pass sharpens the image but converges slowly and occasionally and get better results compared to SDD. SRDD (SDD + MR): our confidence-aware refinement masks only low-confidence tokens. Over five iterations (columns 1,2,3,5 from left to right), it progressively recovers fine boundaries (goose neck, toucan beak), restores textures (poodle fur), and suppresses background noise, ultimately producing the sharpest, mos̶

**Classifier free guidance** Figure 8 evaluates the FID Heusel (2017) and IS Salimans (2016) obtained by the VAR, SDD, and two enhanced SDD that include Simple resampling (SR) and Token Resampling (MR), across a range of cfg scales. Moderate guidance is optimal for SDD. FID decreases

Figure 8: **Effect of cfg:** We present the effect of cfg on FID and IS.

monotonically from cfg 1 to cfg 5, reaching 17.99 on MINIIMAGENET. IS peaks at simultaneously at 63.28. Beyond cfg 5 both metrics plateau, mirroring the saturation behaviour reported for discrete diffusion models Schiff (2024). VAR collapses under strong guidance, leading to an increase in FID (20 → 27) and a decrease in IS (60 → 51) as cfg grows. We attribute this to over-conditioning: without an explicit noise schedule, large cfg values suppress token-level entropy and hinder the performance. Both SR and MR yield uniformly lower FID than SDD for cfg 1–4 and maintain near-optimal performance for cfg values 6–10. Iterative feedback re-injects stochasticity after each guidance pass, preventing the over-conditioning collapse predicted by theory Zhang (2024). These results validate the diffusion-theoretic interpretation of the Markovian factorisation.

**Distillation of VAR variants** Large Consistency Models (LCM) Luo et al. (2023b) demonstrate that a diffusion teacher can be distilled into a student that samples in fewer denoising steps. Since SDD is likewise a multi-scale generative process, we ask an analogous question: Can we remove intermediate scales without sacrificing realism?

Starting from pre-trained checkpoints of SDD on MiniImageNet, we fine-tune each model with the same cross-entropy objective but only on a subset of its original scales. Concretely, the full schedule $\{1, 2, 3, 4, 5, 6, 8, 10, 13, 16\}$ is progressively pruned to $\{1, 3, 5, 8, 13, 16\}$ and then to $\{1, 5, 8, 13, 16\}$. We always retain the highest two scales 13 and 16 because they encode high-frequency details that are irreplaceable in practice. Skipping every second scale (schedule 1-3-5-8-13-16) increases FID by only +0.02 (from 17.99 → 18.01) and leaves IS unchanged (61.98 → 62.01) at cfg = 5.0, confirming that early-stage redundancy. More aggressive pruning three consecutive early scales (1-5-8-13-16) yields a moderate FID of 19.48 and an IS of 61.99.

Like diffusion models, SDD can be time-compressed by pruning early coarse scales while preserving the final high-frequency stages. A 6-scale student (1-3-5-8-13-16) achieves a similar FID/IS as the 10-scale teacher, cutting inference cost by 20% without retraining from scratch. The SDD achieves a 1.75× speedup and a 3× reduction in memory usage. Moreover, incorporating scale distillation further improves inference latency and reduces the memory footprint compared to the original VAR. More pruning comparisons are shown in the supplementary

Table 3: **Ablation study across datasets:** SR: Simple Resampling. MR: Mask Resampling. cfg: Optimized Classifier-Free Guidance.

| Method | Conditional | | | | Unconditional | | | |
|---|---|---|---|---|---|---|---|---|
| | MiniImageNet | | SUN | | FFHQ | | AFHQ | |
| | FID(↓) | IS(↑) | FID(↓) | IS(↑) | FID(↓) | IS(↑) | FID(↓) | IS(↑) |
| VAR | 21.01 | 59.32 | 15.72 | 16.19 | 19.23 | 3.09 | 14.74 | 9.92 |
| SDD | 18.03 | 60.99 | 15.29 | 16.23 | 18.89 | 3.05 | 14.03 | 9.97 |
| SDD + cfg | 17.99 | 62.28 | 14.31 | 17.05 | 18.89 | 3.05 | 14.03 | 9.97 |
| SDD + cfg + SR | 16.82 | 63.28 | 14.01 | 17.51 | 17.62 | 3.89 | 13.52 | 9.66 |
| **SDD + cfg + MR: SRDD** | **16.76** | **63.31** | **13.26** | **17.97** | **17.37** | **4.05** | **13.14** | **10.09** |

## 4.3 ZERO-SHOT PERFORMANCE

Following the evaluation protocol of RePaint Lugmayr et al. (2022), we assess in-painting, out-painting, and super-resolution without task-specific

Table 2: **Zero-shot Performance:** We evaluate the zero shot performance on image reconstruction tasks

| Method | Inpaint | | Outpaint | | SR | |
|---|---|---|---|---|---|---|
| | LPIPS↓ | FID↓ | LPIPS↓ | FID↓ | PSNR↑ | SSIM↑ |
| VAR | 0.26 | 29.92 | 0.48 | 54.01 | 18.01 | 0.403 |
| **SDD** | **0.23** | **28.79** | **0.46** | **52.63** | **18.06** | **0.411** |

fine-tuning. A set of 300 validation images is sampled from AFHQ validation set. For the first two tasks, we reuse the publicly released masks of Lugmayr et al. (2022); Table 2 compare the VAR with SDD across four metrics. SDD consistently outperforms the baseline: In-painting. LPIPS drops from 0.26 to 0.23 and FID from 29.92 to 28.79. Out-painting. Similar gains are observed with LPIPS $0.48 \rightarrow 0.46$ and FID $54.01 \rightarrow 52.63$. Super-resolution. SSIM rises from 0.403 to 0.411,dB, while PSNR improves from 18.01 to 18.06. The qualitative comparison is shown in Figure 1.

## 4.4 ABLATION STUDY

To disentangle the impact of the Markovian attention scheme, optimal cfg and the two resampling strategies shown in Secs. 3.3, we conduct ablations on all the benchmarks. Quantitative numbers are summarised in Table 3, while Figure 9 visualises an example for every setting.

Replacing causal masking with Markovian masking yields a consistent reduction in memory cost and improves visual quality on all four benchmarks. For instance, FID drops from 21.01 to 18.03 on MiniImageNet ($-2.98$, $\approx 14\%$ rel-

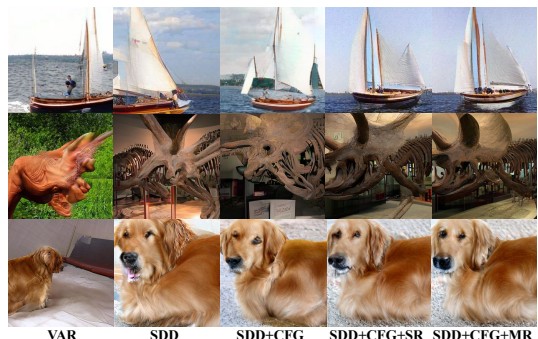

VAR    SDD    SDD+CFG    SDD+CFG+SR    SDD+CFG+MR

Figure 9: **Ablation Study:** Effect of different components of SRDD on performance.

ative), while IS rises from 59.32 to 60.99. Qualitatively (Fig. 9 (a)→(b)), SDD sharpens object boundaries and suppresses artifacts, which confirms that conditioning each scale only on its immediate predecessor is superior for high-quality synthesis, as all the unwanted low-frequency information is discarded in the Markovian style of SRDD. Further, best cfg 5.0 leads to improvement in visual result as shown in (Fig. 9 (b)→(c)).

We perform simple resampling at each scale. This refinement step recovers high-frequency details: FID is reduced by another 1.17 on MiniImageNet, and IS jumps to 63.28. Fig. 9 (c)→d shows crisper textures (e.g. sails and fur) and reduce the artifacts further. Replacing SR with our token-level mask resampling yields the best overall scores on *all* datasets. Relative to the VAR, FID improves by 20.2% on MiniImageNet (21.01→16.76) and 15.6% on SUN (15.72→13.26), while IS gains range from +6.7% to +31.1%. Notably, unconditional FFHQ reaches an IS of 4.05. Figure 9 (e) illustrates that MR selectively sharpens salient regions(the boat's rigging, the dog's face and the body) without introducing over-sharpening artifacts.

## 5 CONCLUSION

We revisited Visual Autoregressive Generation (VAR) through the lens of discrete diffusion and showed that its Markovian variant, SDD, is mathematically equivalent to a structured discrete diffusion process. This perspective explains the bridge between AR transformers and diffusion models, removes inefficiencies of causal conditioning, and enables principled use of diffusion techniques such as classifier-free guidance, token resampling, and scale distillation. Empirically, SDD achieves faster convergence, lower inference cost, and improved zero-shot performance across multiple benchmarks while retaining strong scaling properties. We believe this diffusion-based reinterpretation of VAR provides both theoretical clarity and practical efficiency, opening new directions for scalable and unified visual generation.

## 6 ETHICS STATEMENT

This work studies generative modeling from a theoretical and methodological perspective. All datasets used (Mini-ImageNet, SUN397, FFHQ, AFHQ) are publicly available and widely adopted in research, involving no human subjects or private data. While generative models may be misused to create harmful content, our contributions are intended solely to advance scientific understanding and efficiency of visual generation. We declare no conflicts of interest, and all results are reproducible with the code and checkpoints that will be released.

## 7 REPRODUCIBILITY STATEMENT

We have taken steps to ensure reproducibility of our results. The datasets are publicly available and described in the appendix. Model architecture, training details, and hyperparameters are provided in Section 4 and Appendix. We report all experimental protocols, ablations, and evaluation metrics. Code, pretrained checkpoints, and instructions to reproduce our results will be released upon publication.

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

APPENDIX

## A  DATASETS AND METRICS

We benchmark on two class–conditional datasets. **Mini-ImageNet** Dhillon et al. (2019), containing 50,000 training and 10,000 validation images. **SUN397** Herranz et al. (2016) comprises 108,753 images from 397 scene categories. For computational efficiency, we sample a balanced subset of 175 classes, retaining 150 images per class (26,250 images in total). To evaluate class-agnostic synthesis we adopt two face-centric datasets. **FFHQ** Karras et al. (2019) contains 70,000 high–quality human portraits, while **AFHQ** Choi et al. (2020) contains 15,000 animal faces spanning cats, dogs, and wildlife. All images are resized to $256 \times 256$ before training. For the zero-shot analysis, we draw 300 validation samples from AFHQ and reuse the RePaint Lugmayr et al. (2022) masks. To quantify image fidelity and diversity, we generate 5,000 samples per model and evaluate using FID Heusel (2017) and IS Salimans (2016). For zero-shot editing tasks we report: **LPIPS** Zhang et al. (2018) and FID for in/out-painting, and **PSNR** and **SSIM** for super-resolution. Lower values are better for LPIPS and FID, whereas higher is better for IS, PSNR, and SSIM. **Due to computational constraints in our academic setting, all experiments are conducted on moderately sized datasets.**

## B  FULL ALGORITHM OF FORWARD SAMPLING PROCESS IN VAR

Let $f_\theta$ denote the VAR transformer network , let it operate at a scale $n$, with input $i_{n-1}$, at a scale $n$. The inference algorithm of VAR can be described as

$$\mathbf{f_\theta(i_{n-1}, n)} : i_{n-1} \rightarrow h_n \tag{11}$$
$$f_n = f_{n-1} + (h_n)_{\uparrow(N)} \tag{12}$$
$$i_n = (f_n)_{\downarrow(n+1)} \tag{13}$$

Here $\downarrow (n+1)$ denotes downsampling of the output to a scale $n-1$. $N$ is the largest scale in the sampling process. Other details remain the same as described in the section **Rethinking VAR Variants Through the Discrete Diffusion Lens.**

## C  ANALYSIS

**Classifier free guidance**   We observe in Fig. 10 the same guidance trend on the SUN 397 benchmark. **VAR peaks at a very mild scale.** A guidance weight of **cfg = 2** yields its best trade-off (FID $\downarrow$ **15.55**, IS $\uparrow$ **16.76**); any further increase steadily harms generative quality, reaching FID 27, IS 12 at cfg 10. **Markovian factorisation stabilises guidance.** SDD remains flat until cfg 6, and both refinement heads suppress the residual drift. The mask-resampling variant (MR) attains the global optimum at **cfg = 5** with FID $\downarrow$ **13.26** and IS $\uparrow$ **17.97**, while staying within $\pm 0.3$ FID across the whole 1.5–10 range. This robustness removes the need for dataset-specific tuning and further analysis of our diffusion-style interpretation: iterative resampling continually re-injects entropy, offsetting the over-conditioning collapse that plagues the original VAR decoder.

**Distillation of VAR variants**   To further understand the distillation, we consider three more extreme schedules, visualised in Fig. 11.

- **×3 step (1−5−13−16).** Dropping two out of every three scales reduces decoder passes by $2.5\times$ but also degrades quality: FID jumps to 31.21 (cfg 3.5) and 29.83 (cfg 5.0), while IS decreases to $48.25/48.61$. The student fails to reconstruct *both* low-frequency layout and high-frequency details—suggesting that coarse-to-fine refinement needs at least one *intermediate* scales.

- **Early-heavy (1−2−3−4−5−8−16).** Retaining a single high scale pass is insufficient: FID deteriorates to 39.91 (cfg 5.0) and IS collapses to 38.95. Qualitative inspection reveals blurry textures and colour bleeding, indicating that high-frequency content injected at scale 16 cannot overwrite errors accumulated during the densely sampled low-scale stages.

- **Random sparse (`1-4-8-16`).** A non-uniform, randomly spaced schedule performs worst (FID $38.76$, cfg $5.0$). Without a consistent geometric progression, successive decoders operate on feature maps whose receptive fields overlap poorly, breaking the iterative error-correction mechanism that underpins multi-scale generation.

Across *all* settings, the Markovian variant (SDD) remains strictly better than the original VAR, mirroring the trends. Consecutive low-resolution scales are largely redundant, but at least two high-resolution scales are indispensable. A simple distillation of `1-3-5-8-13-16` is therefore near-optimal—cutting inference time by $20\%$ while preserving perceptual quality within $8\%$ of the teacher.

## D  THEORETICAL EQUIVALENCE OF MARKOVIAN VAR AND DISCRETE DIFFUSION

**Definition 1 (Markovian VAR)** *Let $(x_0, \ldots, x_N)$ denote the sequence of discrete latent images (or token grids) from the coarsest scale $1$ to the finest scale $N$, where $x_0$ and $x_N$ denote the original image. For each $n = 1, \ldots, N$, let $M(n)$ be a non-linear deterministic downsampling transition mapping $x_{n+1}$ to the coarser representation $x_n = M(n) \, x_{n+1}$.*

*A* Markovian VAR *model specifies:*

- *a prior $p(x_N)$ over the finest scale;*

- *a final stationary state $x_1$, which is $\langle SOS \rangle$ in our case;*

- *for each $n = 1, \ldots, N$, a conditional distribution $p_\theta(x_{n+1} \mid x_n)$, parameterized by a transformer that is allowed to attend within scale $n$ and only to the immediately coarser scale $n$ (Markovian attention across scales).*

*Generation proceeds by sampling*

$$x_1 \sim p(x_1), \qquad x_{n+1} \sim p_\theta(x_{n+1} \mid x_n) \quad for \ n = 1, \ldots, N.$$

**Definition 2 (Discrete diffusion model)** *Let $(x_0, \ldots, x_N)$ be a sequence of discrete states and let $Q_n$ be an absorption transition matrix. Define a forward Markov chain $q(x_{1:N} \mid x_0)$ by*

$$q(x_n \mid x_{n-1}) = x_{n-1} Q_n, \qquad n = 1, \ldots, N,$$

*i.e., $x_n$ is obtained by transitioning $x_{n-1}$ through $Q_n$.*

*Given a prior $p(x_N)$ and reverse kernels $p_\theta(x_{n-1} \mid x_n)$, the associated* discrete diffusion model *is the joint distribution*

$$p_\theta(x_0, \ldots, x_N) = p(x_N) \prod_{n=1}^{N} p_\theta(x_{n-1} \mid x_n).$$

**Proposition 1 (Markovian VAR is a discrete diffusion model)** *Consider a Markovian VAR model as in Definition 1 and the discrete diffusion model in Definition 2. Assume:*

1. **Deterministic forward chain.** *For all $n = 1, \ldots, N$, $q(x_n \mid x_{n+1}) = M(n) \, x_{n+1}$, i.e., the forward process is the fixed non-linear downsampling transition induced by $M(n)$.*

2. **Shared reverse kernels.** *The reverse kernels $p_\theta(x_{n-1} \mid x_n)$ in the diffusion model coincide with the Markovian VAR conditionals $p_\theta(x_{n+1} \mid x_n)$.*

3. **Markovian VAR training loss.** *The Markovian VAR is trained with the token-wise cross-entropy loss*

$$\mathcal{L}_{\mathrm{o}}(\theta) = -\sum_{n=1}^{N} \mathbb{E}_{q(x_N)} \, \mathbb{E}_{q(x_n \mid x_N)} \log p_\theta(x_{n+1} \mid x_n),$$

*where $q(x_n \mid x_N)$ is obtained by running the deterministic forward chain from $x_N$ down to scale $n$.*

*Then:*

1. *The joint distribution of the Markovian VAR model is exactly a discrete diffusion model with forward kernel $q$ and reverse kernel $p_\theta$:*

$$p_\theta(x_0, \ldots, x_N) = p(x_N) \prod_{n=1}^{N} p_\theta(x_{n-1} \mid x_n).$$

2. *The discrete diffusion variational bound*

$$\mathcal{L}_{\mathrm{v}} = \mathbb{E}_{q(x_0)} \left[ \underbrace{D_{\mathrm{KL}}\big(q(x_T \mid x_0) \,\|\, p(x_T)\big)}_{L_T} + \sum_{t=2}^{T} \mathbb{E}_{q(x_t|x_0)} \underbrace{D_{\mathrm{KL}}\big(q(x_{t-1} \mid x_t, x_0) \,\|\, p_\theta(x_{t-1} \mid x_t)\big)}_{L_{t-1}} \right.$$

$$\left. - \underbrace{\mathbb{E}_{q(x_1|x_0)}[\log p_\theta(x_0 \mid x_1)]}_{L_0} \right]$$

*satisfies*

$$\mathcal{L}_{\mathrm{v}}(\theta) = \mathcal{L}_{\mathrm{o}}(\theta) + C,$$

*where $C$ is a constant independent of $\theta$.*

*In particular, minimizing the Markovian VAR training loss is exactly equivalent to maximizing the discrete diffusion ELBO. Thus, Markovian VAR is a discrete diffusion model in the D3PM sense with deterministic forward transitions.*

## E  LIMITATIONS

While our results are promising, the proposed SRDD framework still has several practical and scientific limitations that future work should address.

- **Compute budget.** All experiments were run on just 4 NVIDIA A6000 GPUs for 200 epochs. This constraint forced us to use reduced versions of the training datasets and limited the largest model size we could explore. Larger-scale training might uncover different failure modes or reveal further gains that we could not test in our setting.

- **Dataset scope.** We evaluate on four medium-scale image collections—Mini-ImageNet, SUN397 (subset), FFHQ, and AFHQ. We cover only $256 \times 256$ resolution and a modest range of visual diversity. Consequently, it remains unclear how SRDD performs on very high-resolution images, highly complex scenes (e.g., ImageNet-1k, COCO), or video.

- **Codebook expressiveness.** Like VAR, we rely on a single VQ-VAE codebook. Although efficient, this discrete bottleneck can limit fine detail and color accuracy compared with continuous-latent diffusion models.

## F  FUTURE WORK

Although *Scalable Refinement with Discrete Diffusion* (SRDD) already improves upon VAR across several axes and finds a closer interpretation with discrete diffusion models, we see at least four promising directions for further research:

- **Larger-scale pre-training and scaling laws.** Our results hint that SRDD follows the same parameter–quality trend observed in VAR. A systematic scaling over wider model sizes, sequence lengths, and token vocabularies could reveal precise scaling laws, guiding practitioners toward the most compute-efficient regimes Kaplan et al. (2020); Hoffmann et al. (2022).

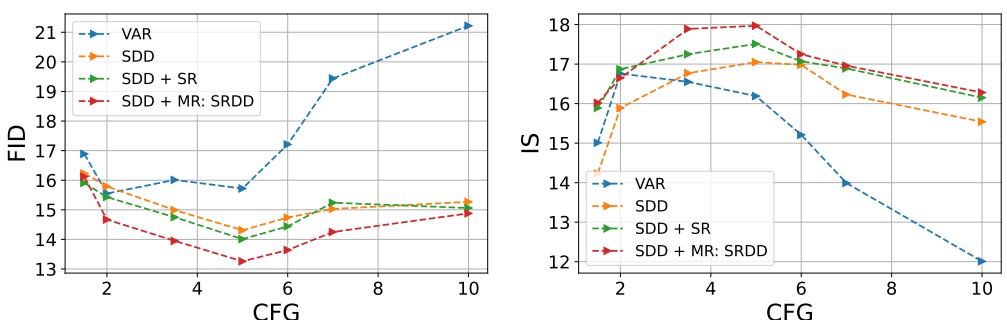

Figure 10: **Effect of cfg on SUN397 Dataset:** We Present the effect of cfg on FID and IS Score

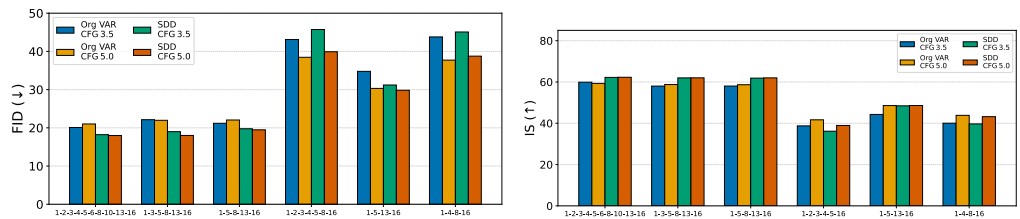

Figure 11: **Effect of distillation on reducing the number of scales**

- **Learned resampling policies.** The current MR strategy uses a fixed probability threshold. Replacing this hand-tuned rule with a small policy network—trained to predict which tokens to resample given the decoder's uncertainty—might yield further gains while cutting the number of refinement passes.

- **Continuous–discrete hybrid diffusion.** SRDD operates in a purely discrete latent space; continuous-time diffusion models excel in capturing fine textures. A hybrid pipeline that first runs SRDD at coarse scales and then applies a lightweight continuous decoder (e.g. a UNet) for final touch-ups could combine the speed of SRDD with the photorealism of continuous diffusion Song et al. (2020); Peebles & Xie (2022).

- **Leveraging advances in discrete diffusion theory.** We showed that the Markovian variant of VAR is theoretically and empirically equivalent to a discrete diffusion process. As the community uncovers new principles—e.g., refined noise schedules, tighter ELBO bounds, or more stable discretisations—these insights can be transferred to SRDD, offering a low-cost pathway to inherit future breakthroughs in discrete diffusion.

# G LLM USAGE

We acknowledge that Large Language Models (LLMs) were used to assist with refining the clarity of the writing in this manuscript.

**DiT − L/2**

**VAR**

**OURS: SRDD**

Figure 12: **Qualitative Comparison of DiT-L/2, VAR and Ours: SRDD; We do not compare with LDM because LDM model didn't converage**

**LDM**

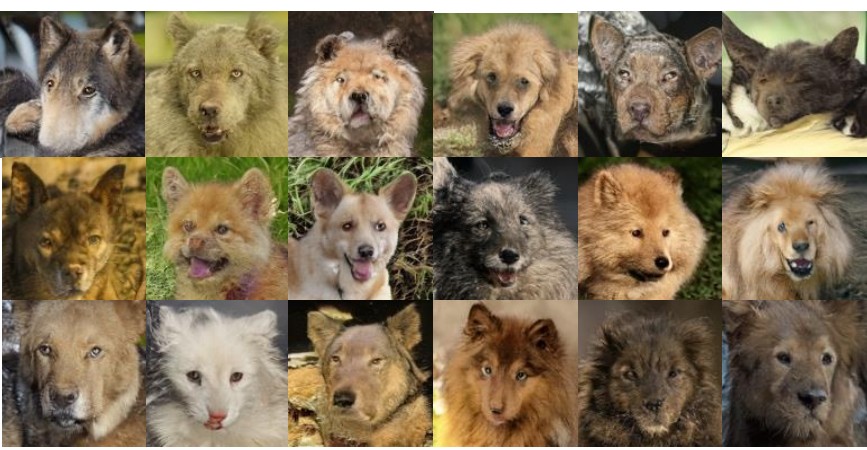

**VAR**

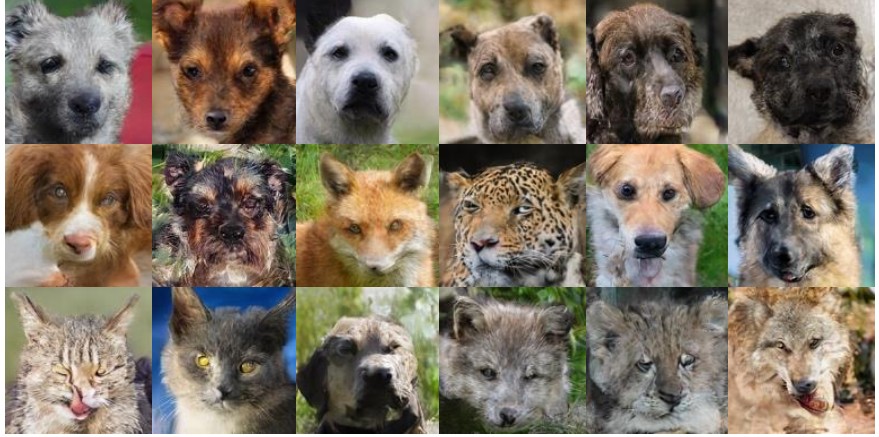

**OURS: SRDD**

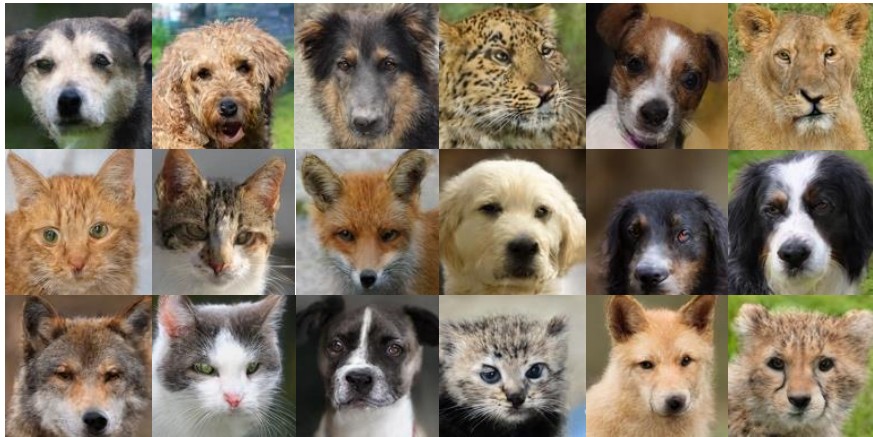

Figure 13: **Qualitative Comparison on AFHQ Datasets, LDM, DiT-L/2, VAR and SRDD**: DiT-L/2 didn't converage on AFHQ Datatsets

**LDM**

**DiT − L/2**

**VAR**

**OURS: SRDD**

Figure 14: **Qualitative Comparison of FFHQ Datasets LDM, DiT-L/2, VAR and SRDD**

**LDM**

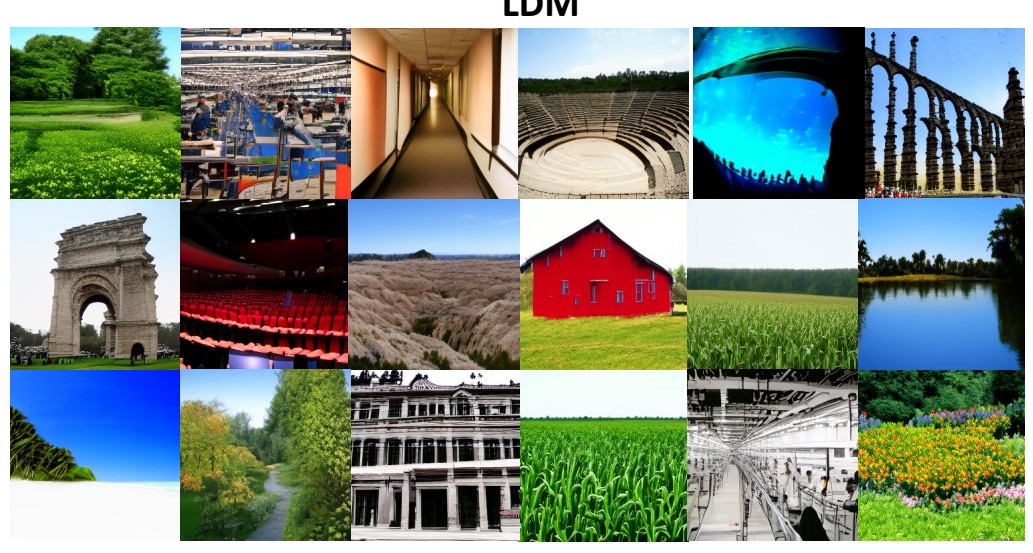

**VAR**

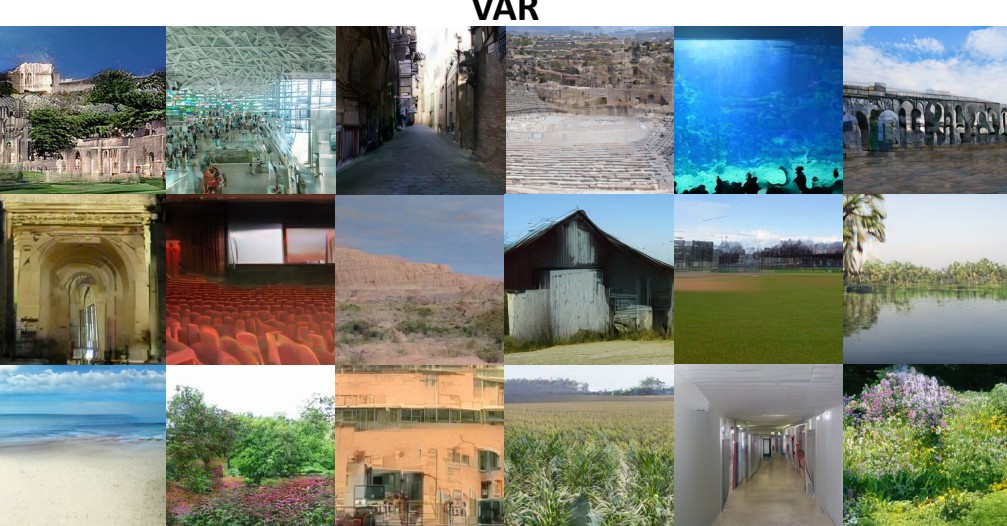

**OURS: SRDD**

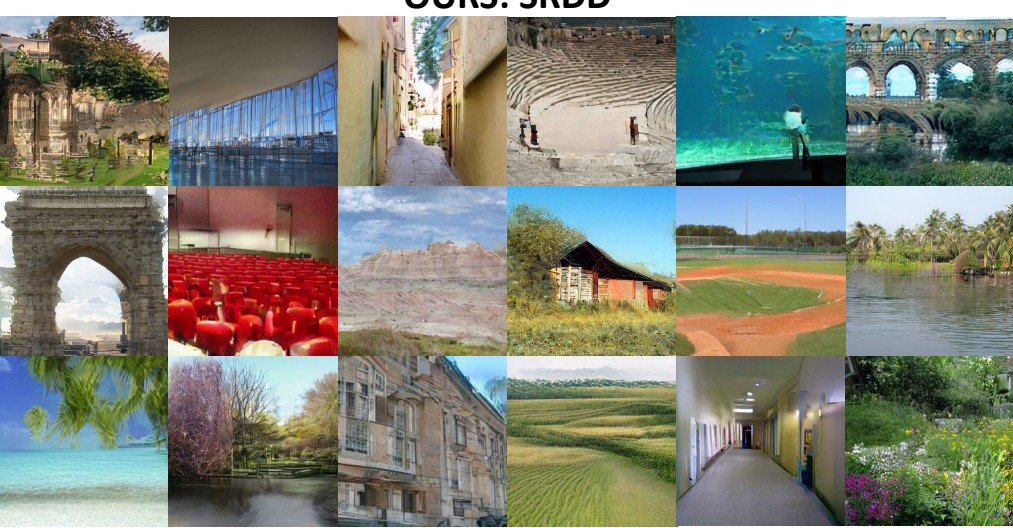

Figure 15: **Qualitative Comparison on SUN397 Datasets, LDM, DiT-L/2, VAR and SRDD**

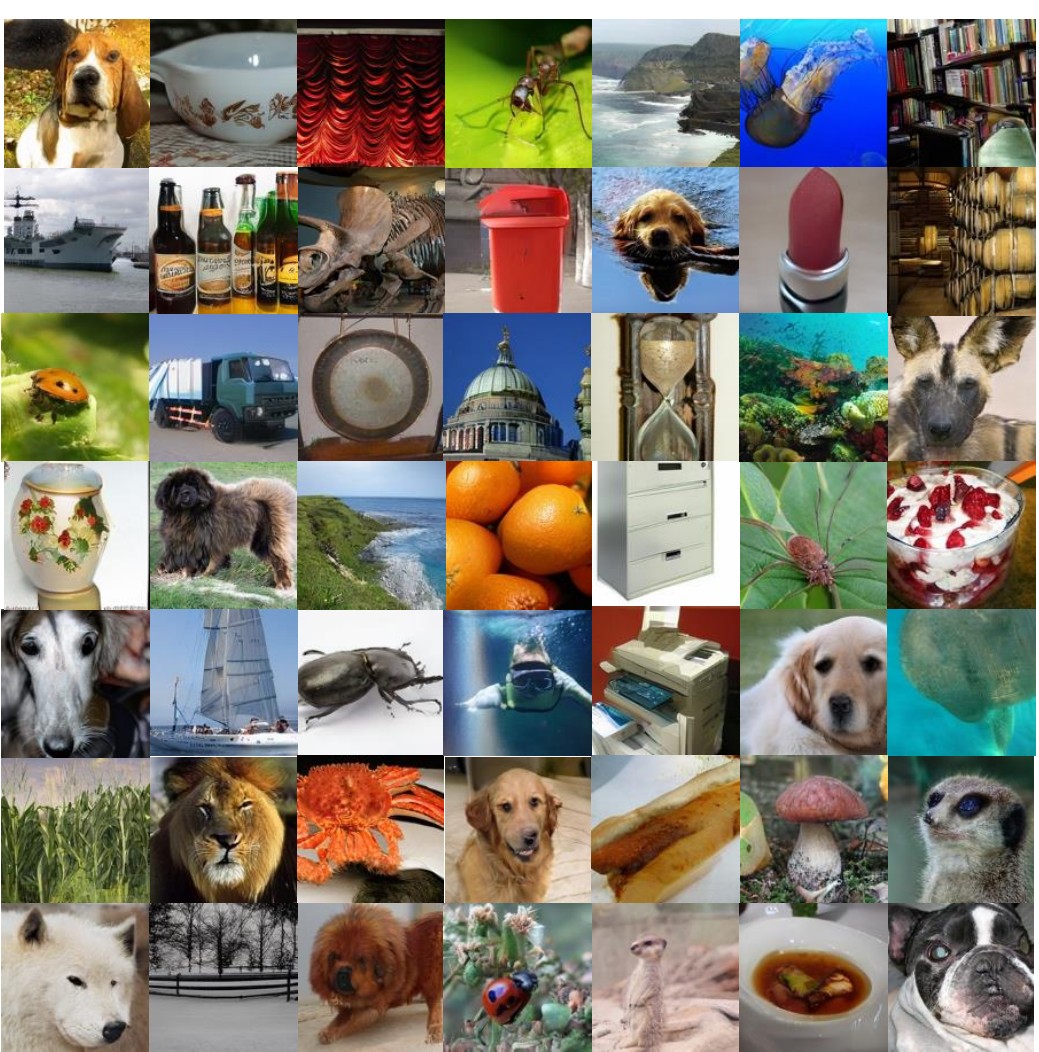

Figure 16: Non-curated example images generated by the proposed SRDD approach for the MiniImagenet Dataset

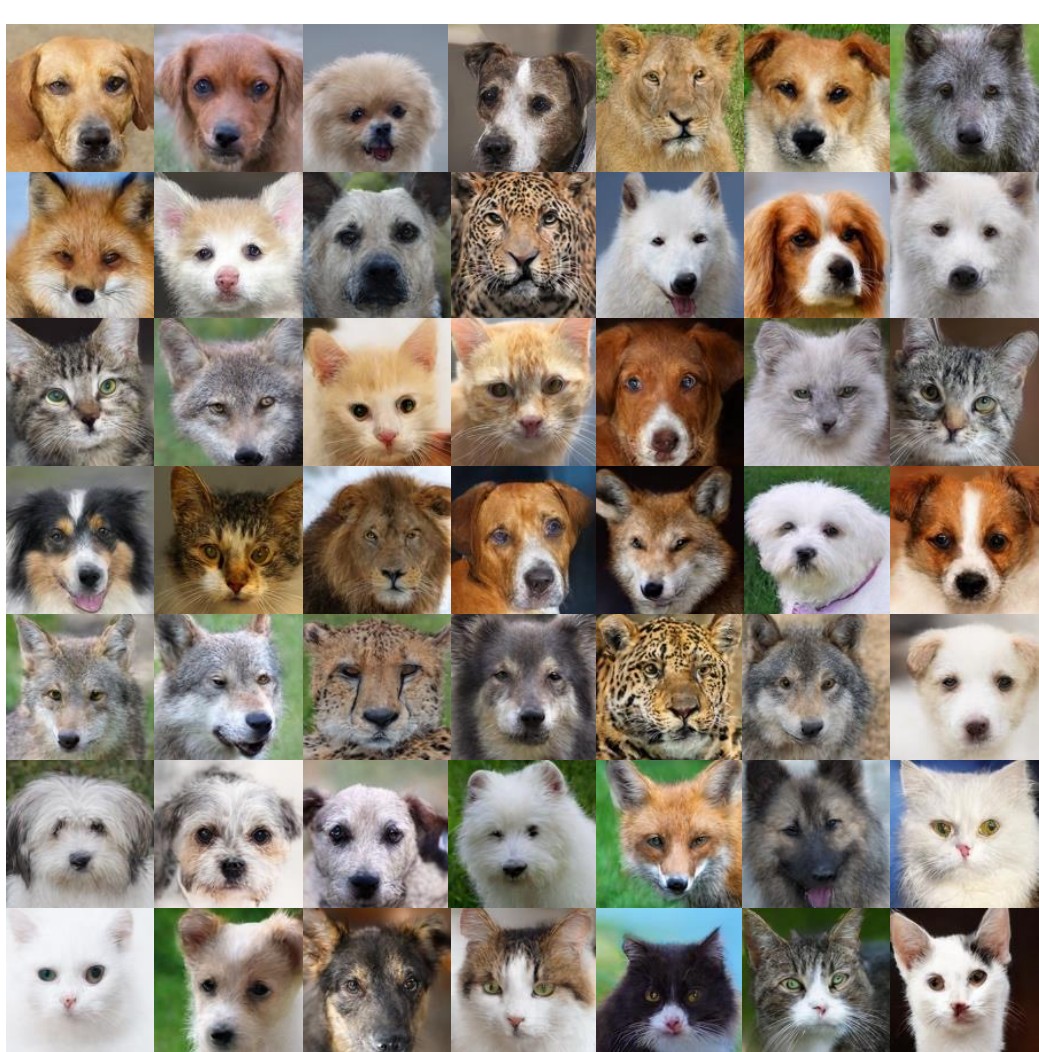

Figure 17: Non-curated example images generated by the proposed SRDD approach for the AFHQ Dataset

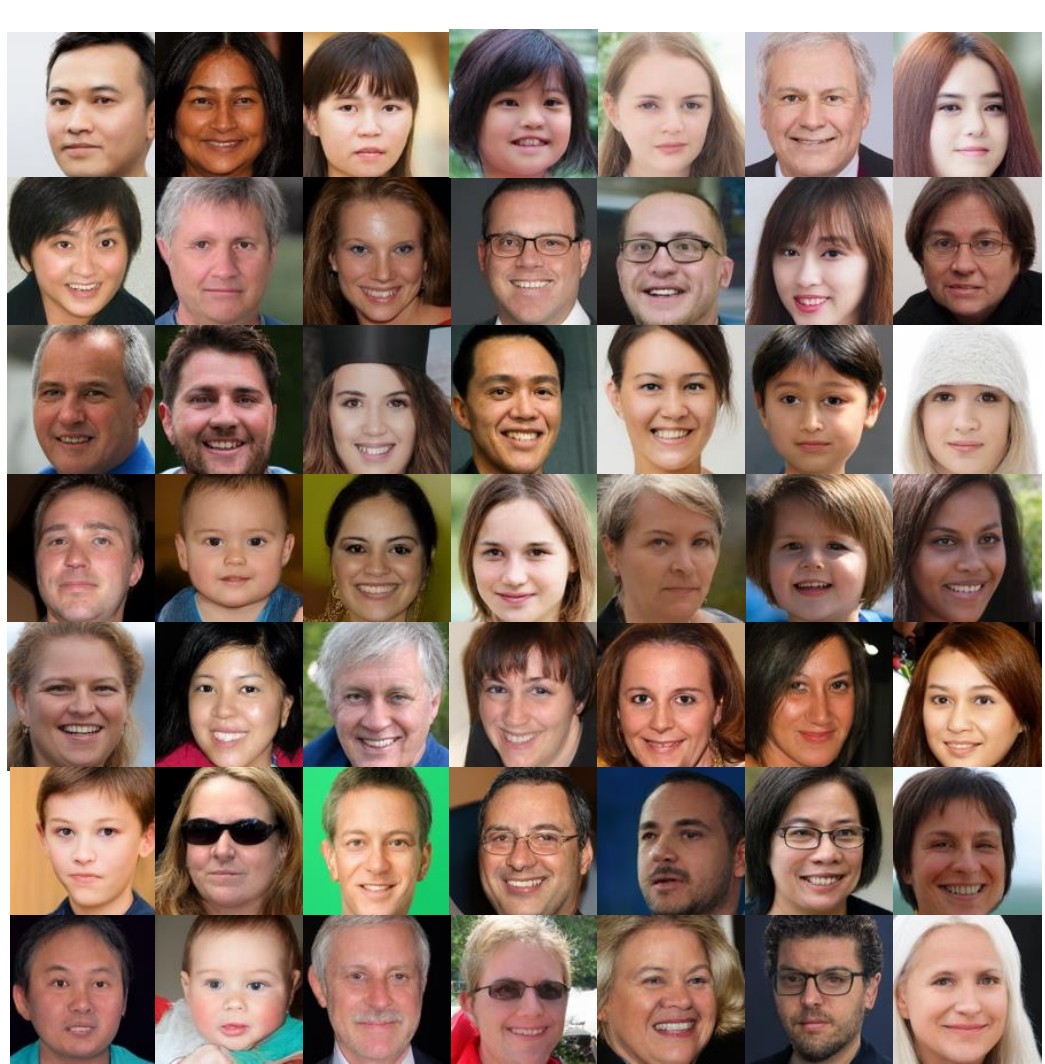

Figure 18: Non-curated example images generated by the proposed SRDD approach for the FFHQ Dataset

