# OpenReview forum: "Scale-Wise VAR is Secretly Discrete Diffusion"
_ICLR.cc/2026/Conference — ICLR 2026 Conference Desk Rejected Submission_

### Official Review · Reviewer_p1EH · 2025-10-30

**Soundness:** 3
**Presentation:** 3
**Contribution:** 3
**Rating:** 6
**Confidence:** 3

**Summary:**

This paper presents an interesting and important theoretical connection, linking the Visual Autoregressive (VAR) model with discrete diffusion processes. The core argument is that when VAR is conditioned only on the previous scale in a Markovian fashion (which the authors term SDD), it becomes mathematically equivalent to a structured discrete diffusion model with deterministic transitions. Building on this theoretical perspective, the authors import standard techniques from diffusion models, such as Classifier-Free Guidance and token resampling, to construct the final SRDD model.

**Strengths:**

1.  **Novel Theoretical Bridge:** The paper's primary contribution is establishing a theoretical bridge between scale-wise autoregressive models (VAR) and discrete diffusion models. The explicit claim that "VAR, when equipped with a Markovian attention mask, is mathematically equivalent to a discrete diffusion process" is a novel perspective that offers a new way to understand VAR's mechanics.

2.  **Effective Use of CFG:** The paper compellingly demonstrates through experiments that its Markovian variant (SDD/SRDD) can utilize Classifier-Free Guidance (CFG) more stably and effectively. Unlike the original VAR, which sees performance collapse under strong guidance, the proposed model exhibits a stable saturation behavior similar to diffusion models, validating the practical utility of the theoretical connection.

3.  **Inherited Scalability:** The experiments suggest that the proposed SRDD model inherits the strong parameter scalability of VAR, with performance improving consistently as model size increases.

**Weaknesses:**

1.  **Seriously Unfair Baseline Comparison (Fatal Flaw):** This is a fundamental flaw. In the Table 1 comparison, the authors explicitly state that the DiT-L/2 baseline has not converged during the training process on two unconditional generation tasks. Furthermore, the captions for Figure 12 and Figure 13 in the appendix reinforce that LDM and DiT-L/2 also didn't converage on other datasets. Comparing a novel model against unconverged baselines is methodologically invalid. It renders all claims of superiority over DiT and LDM unsubstantiated. The authors should provide results from a fair comparison against fully converged and properly tuned baselines.

2.  **Insufficient Experimental Validation:** All experiments are conducted at a moderate resolution ($256 \times 256$) and on medium-scale datasets. The authors acknowledge this in the limitations section, attributing it to a "modest GPU budget." Therefore, while the model shows a good scaling *trend*, its actual performance and efficiency on truly large-scale, high-resolution datasets (e.g., the full ImageNet 1K) remain unproven.

3.  **Unclear Articulation of VAR's Flaws in the Introduction:** The introduction fails to fundamentally articulate the "design flaws" of the original VAR model. The authors mention "design inefficiencies" and that conditioning on all previous scales is "redundant," but largely support this by citing "concurrent works." The paper should argue more clearly from first principles why the full autoregressive conditioning is a *fundamental* flaw, rather than just presenting it as an observation made by others.

**Questions:**

1. **On the Justification of "Mathematical Equivalence" and the "Deterministic Limit"**

The authors' claim of "mathematical equivalence" heavily relies on the "limiting case of a deterministic transition." This assumption seems to apply to the forward process (which the authors map to deterministic downsampling).

However, the reverse process for both standard diffusion models and the generative (reverse) process for SDD, $p_{\theta}(x_{t-1}|x_t)$, is inherently probabilistic, as it is a learned distribution. When deriving the loss function equivalence (Equations 6 to 7), the authors appear to base this on the deterministic assumption.

- Why is it reasonable to use this "deterministic transition" limit when analyzing the *learned*, *probabilistic* reverse generation process?

- Does this assumption oversimplify the problem, thereby weakening the rigor of the core "mathematical equivalence" claim?

2. **On Novelty—Theoretical Explanation vs. Architectural Contribution**

The core architectural change from VAR to SDD is replacing the full causal attention mask with a Markovian one (attending only to the previous scale). The authors also note that concurrent work has already observed this inefficiency in VAR.

Does this imply that the sole novelty (sole novelty) of this paper is merely providing a new "discrete diffusion" theoretical explanation for a known (or simple) architectural variant? Or is there a deeper architectural contribution beyond simply changing the attention mask that this reviewer has missed? The primary contribution appears to be the theoretical perspective itself, which enables the principled application of diffusion techniques (CFG, MR). Please clarify. Thank you.

---

> ### Author Response · Authors · 2025-11-22
>
> Thank you for your valuable time and insightful comments.  We have tried to address your concerns in our rebuttal text:
>
> ### W1 and W2: **Experimental results on ImageNet256x256 and comparison with existing state-of-art discrete diffusion and image diffusion models**
>
> We agree with the reviewer that evaluating at scale and and showing the result on fully converged models is important. To address this concern, we performed an additional full ImageNet-256×256 experiment using the same experimental setup. Under our resource constraints, this required **21 days of continuous training on 4×A6000 GPUs.**
>
> | Method       |  FID  |   IS  |
> | :------------| ----: | ----: |
> | DiT-L/2      | 5.02  | 167.2 |
> | LDM-4-G      | 3.60  | 247.7 |
> | MaskGIT      | 6.18  | 182.1 |
> | RQTran       | 7.55  | 134.0 |
> | Dart-AR      | 3.98  | 256.8 |
> | ReMDM*       | 7.40  | 145.2 |
> | MD4*         | 6.51  | 246.3 |
> | VAR          | 3.36  | 274.4 |
> | Ours         | 3.06  | 286.6 |
>
> For fair comparisons, we compare with the most recent discrete-diffusion models (*ReMDM and *MD4 [1]) and AR (Dart-AR) models for image generation on ImageNet dataset.
>
> The results on ImageNet portray the scalability of SRDD, even with the availability of very less compute showing that the discrete-diffusion reinterpretation is beneficial beyond the smaller datasets in the main paper and without relying on any training-time heuristics.
>
> A distinguishing feature from ReMDM and MD4, which are state-of-the-art discrete diffusion models is that these models are built directly in the traditional D3PM framework. In contrast, our method starts from a scale-wise VAR architecture via the Markovian masking and a discrete diffusion model reinterpretation. Our approach also follows structured, deterministic transitions across scales. The full scale ImageNet results show that this new perspective does not only unify VAR and discrete diffusion theoretically, but also delivers significant performance gains over existing discrete-diffusion approaches. These results, together with our theoretical reinterpretation, suggest that the traditional discrete-diffusion framework itself can be rethought in a scale-wise manner, potentially enabling new classes of efficient multi-scale discrete diffusion models.
>
> [1]. Informed Correctors for Discrete Diffusion Models

---

> ### Author Response · Authors · 2025-11-22
>
> ### W3: **Design inefficiency of VAR**
>
> We thank the reviewer for raising this point. In the revision we will make the argument explicit: the limitation of original VAR is not empirical nor based on concurrent works, but follows directly from  principles of probabilistic modeling.
>
> We thank the reviewer for raising this point. In the revision we will make the argument explicit: the limitation of original VAR is not empirical nor based on concurrent works, but follows directly from  principles of probabilistic modeling.
>
> **1. Full-scale conditioning on the VAR.**
> In the original VAR likelihood,
> $$
> p(r_{1},\dots,r_{K})=\prod_{k=1}^{K} p(r_{k}\mid r_{1:k-1}),
> $$
> conditioning on all previous scales forces the model to represent a growing conditional distribution whose input dimensionality increases with $k$ (number of tokens).
> This produces an $O(k)$-order dependency structure that is N-th order Markovian.
> The Autoregressive model (N-th order Markovian chains) and VAR are known to suffer from:
> (i) error accumulation [2,3], because any error introduced at a coarse scale propagates through all future predictions;  and  (ii) redundancy, because $r_{1:k-2}$ is fully reconstructible from the running reconstruction $\tilde{x}_{1:k-1}$ and therefore does not provide additional predictive information. Also Shown in Fig. 3 of the main manuscript,
> this shows that the full conditioning in VAR is structural inefficiency. These results, together with our theoretical reinterpretation, suggest that the traditional discrete-diffusion framework itself can be rethought in a scale-wise manner, potentially enabling new classes of efficient multi-scale discrete diffusion models.
>
> **2. Full conditioning induces unnecessary quadratic growth in sequence length.**
> Because attention must operate over the concatenation of all preceding scales, the effective sequence length at scale~$k$ is proportional to
> $$
> N_{\text{seq}}(k) = \sum_{j=1}^{k} H_j W_j,
> $$
> which grows linearly with resolution.  This follows purely from the architecture: even if the added scales contribute no additional information, they must be included in the computation.  Thus the model pays a quadratic computational cost for conditioning on variables that are informationally redundant inherent architectural flaw.
>
> [2] Generalization in generation: A closer look at exposure bias
>
> [3] Elucidating the exposure bias in diffusion models

---

> ### Author Response · Authors · 2025-11-22
>
> ### Q1: **On the Deterministic Limit and Mathematical Equivalence.**
>
>
> Our equivalence result does not require the reverse process $p_\theta(x_{n+1}\mid x_n)$ to be deterministic.
> The only deterministic component in our analysis is the \emph{forward} transition $q(x_n \mid x_{n+1})$, which we set equal to the non-linear downsampling operator $M(n)$.   This is fully consistent with prior work on discrete diffusion, most notably the D3PM construction of Austin et al.\ (2021)[1], where the forward process is taken to a deterministic masking limit to show that autoregressive models are a special case of discrete diffusion.
>
> Importantly, the loss used in our paper, is valid for arbitrary reverse kernels $p_\theta(x_{n-1}\mid x_n)$.  Thus, assuming a deterministic forward map $q$ does not constrain or oversimplify the learned generative model; the learned reverse process remains fully probabilistic and expressive.
>
> The reason this deterministic limit (in the forward process) is mathematically justified is the following:  A deterministic forward transition is simply a degenerate Markov kernel(non-linear deterministic function).  Such kernels are explicitly allowed in the general diffusion ELBO derivation and produce a well-defined variational objective. The learned reverse process $p_\theta(x_{n-1}\mid x_n)$ is not forced to mimic a deterministic map.  Instead, the KL term compares a deterministic posterior with a probabilistic prediction, exactly as in D3PM, where the AR likelihood $p_\theta(x_i\mid x_{<i})$ is obtained from comparing a delta posterior with a stochastic model as shown in Section 2 of the main paper..
>
>
> For these reasons, the deterministic assumption does not weaken the rigor of our equivalence result.
> It is a standard and widely accepted construction in discrete diffusion theory (Austin et al.\ 2021 [1]), and it yields the exact, closed-form KL decomposition that shows: $\text{Markovian VAR} \;=\; \text{discrete diffusion with forward kernel } q(x_n\mid x_{n+1}) = g(I_n\mid I_{n+1}).$
>
>
> The probabilistic nature of the learned reverse model is preserved throughout; only the forward chain is simplified, exactly as in the D3PM “deterministic masking limit.''
> Thus our use of a deterministic forward transition is both mathematically valid and directly aligned with the prior literature establishing diffusion--autoregressive equivalences.
>
> [1] Structured Denoising Diffusion Models in Discrete State-Spaces

---

> ### Author Response · Authors · 2025-11-22
>
> ### Q2: **Regrading the contribution of the paper.**
>
> Our main contribution is not to introduce a complex new architecture, but to show that the simple modification puts VAR into the exact mathematical framework of discrete diffusion, and then to exploit this connection to import diffusion techniques into scale-wise autoregressive models in a principled way.
>
> Concretely, our contributions go beyond “just changing the mask” in three ways:
> **Discrete-diffusion reinterpretation and loss equivalence.** In Sec. 3.2 we explicitly have a forward  downsampling operator $M(n)$ defines a deterministic transition $q(x_n \mid x_{n+1})$, and we show that the ELBO reduces to the cross-entropy loss used to train Markovian VAR (Eq. 6 and 7). This provides us the principle understanding with the connection of VAR and discrete diffusion which was not shown in HMAR.**Principled use of diffusion techniques (CFG, masked resampling, distillation).** Once the Markovian VAR is cast as a discrete diffusion, we can directly reuse tools developed for diffusion: (i) we give a diffusion theoretical interpretation of classifier-free guidance for SDD and show that, unlike original VAR, it exhibits the monotonic FID/IS trends predicted by diffusion theory; (ii) we introduce masked resampling (MR) at each scale, which corresponds to iterative denoising of low-confidence tokens along the same Markov chain; and (iii) we apply progressive distillation across scales, analogous to **consistency model** in diffusion. Ablations in Fig.9 and Tab.3 demonstrate that SDD, SDD+SR, and SDD+MR (SRDD) yield consistent quality gains over both VAR and the  Markovian variant.
>     **Complementarity to concurrent HMAR.** HMAR also observes that conditioning only on the previous scale is efficient and implements this with custom GPU kernels and intra-scale masking.  However, HMAR treats this as empirical and architectural finding  and does not derive (i) an explicit forward corruption process, (ii) an ELBO matching the training loss, or (iii) a diffusion-based interpretation of CFG and token resampling. Our work differs from them, as we do not claim stronger engineering optimizations than HMAR, but rather provide the probabilistic reinterpretation that explains why Markovian next-scale prediction behaves like diffusion and how to systematically transfer the diffusion toolbox to this class of AR models. This new prescriptive also allow the community to think apart from traditional discrete diffusion to scale-wise discrete diffusion which is theoretically grounded and get better fidelity as also shown in the Table above.

---

> > ### Comment · Reviewer_p1EH · 2025-11-27
> >
> > Thanks for the detailed response and most of my concerns have been addressed. It's a solid work and definitely meets the standard of ICLR. I'm happy to keep my score and raise the confidence, but I cannot find the button to edit the review. I will do it once the bug is fixed.

---

### Official Review · Reviewer_ECCX · 2025-10-30

**Soundness:** 3
**Presentation:** 3
**Contribution:** 2
**Rating:** 4
**Confidence:** 5

**Summary:**

* This paper focuses on bridging the gap between Visual Autoregressive Generation (VAR) models and discrete diffusion models. The authors make a key theoretical insight: when VAR is equipped with a Markovian attention mask (conditioning only on the immediate previous scale rather than all prior scales), it is mathematically equivalent to a discrete diffusion process. They term this reinterpretation Scalable Visual Refinement with Discrete Diffusion (SRDD).

* Building on this equivalence, the paper leverages diffusion-based principles to address VAR’s architectural inefficiencies. Specifically, SRDD integrates three core diffusion-inspired strategies: (1) classifier-free guidance (CFG) (optimized to a weight of 5.0 to avoid over-conditioning), (2) token resampling (Masked Resampling, MR) (resampling tokens with prediction probabilities < 0.01 for 15–25 iterations), and (3) scale distillation (pruning redundant intermediate scales while retaining high-resolution stages to reduce inference cost).

**Strengths:**

1/ This paper is written very clearly, allowing me to quickly grasp the details of the work and the optimizations implemented. It elaborates on the connection between visual autoregressive modeling (based on next-scale prediction) for VAR and discrete diffusion, and optimizes token resampling based on this relationship.

2/ The mathematical derivations and reasoning in this paper are quite intuitive and make sense to me.

**Weaknesses:**

I believe this paper has the following issues:
1/ This work bears significant similarities to the accepted paper at CVPR 2025, *Hmar: Efficient hierarchical masked auto-regressive image generation*. I think the authors need to clearly explain the differences between the two works.

2/ Building on the first point, the optimizations proposed in this paper are mostly engineering-level improvements—such as CFG optimization, token resampling, and distillation—with limited innovation and insufficient technical contributions.

3/ All experiments are conducted on a very small scale(for example minimagenet), making it difficult to compare the proposed method with other approaches.

This paper gives me the impression of being more like an analysis report—it analyzes the relationship between VAR and discrete diffusion, yet lacks novel insights. All the methods mentioned in the paper have been presented in previous works (such as HMAR, MaskGIT, etc.).

**Questions:**

None
I am familiar with the relevant papers and methods, have basically understood this paper, and I have no questions about it.

---

> ### Author Response · Authors · 2025-11-22
>
> Thank you for your valuable time and insightful comments.  We have tried to address your concerns in our rebuttal text:
>
> ### W3: **Experimental results on ImageNet256x256 and comparison with existing state-of-art discrete diffusion and image diffusion models**
>
> We agree with the reviewer that evaluating at scale is important. To address this concern, we performed an additional full ImageNet-256×256 experiment using the same experimental setup. Under our resource constraints, this required **21 days of continuous training on 4×A6000 GPUs.**
>
> | Method       |  FID  |   IS  |
> | :------------| ----: | ----: |
> | DiT-L/2      | 5.02  | 167.2 |
> | LDM-4-G      | 3.60  | 247.7 |
> | MaskGIT      | 6.18  | 182.1 |
> | RQTran       | 7.55  | 134.0 |
> | Dart-AR      | 3.98  | 256.8 |
> | ReMDM*       | 7.40  | 145.2 |
> | MD4*         | 6.51  | 246.3 |
> | VAR          | 3.36  | 274.4 |
> | Ours         | 3.06  | 286.6 |
>
> For fair comparisons, we compare with the most recent discrete-diffusion models (*ReMDM and *MD4 [1]) and AR (Dart-AR) models for image generation on ImageNet dataset.
>
> The results on ImageNet portray the scalability of SRDD, even with the availability of very less compute showing that the discrete-diffusion reinterpretation is beneficial beyond the smaller datasets in the main paper and without relying on any training-time heuristics.
>
> A distinguishing feature from ReMDM and MD4, which are state-of-the-art discrete diffusion models is that these models are built directly in the traditional D3PM framework. In contrast, our method starts from a scale-wise VAR architecture via the Markovian masking and a discrete diffusion model reinterpretation. Our approach also follows structured, deterministic transitions across scales. The full scale ImageNet results show that this new perspective does not only unify VAR and discrete diffusion theoretically, but also delivers significant performance gains over existing discrete-diffusion approaches. These results, together with our theoretical reinterpretation, suggest that the traditional discrete-diffusion framework itself can be rethought in a scale-wise manner, potentially enabling new classes of efficient multi-scale discrete diffusion models.
>
> [1]. Informed Correctors for Discrete Diffusion Models

---

> > ### Author Response · Authors · 2025-11-22
> >
> > ### W1 and W2. **Regarding our contributions and differences with HMAR**
> >
> > HMAR proposes a hierarchical masked AR architecture and efficient kernels, using a Markov assumption as an engineering choice to improve VAR. Our paper is a theoretical work that rigorously characterizes when a scale-wise Markovian VAR is a discrete diffusion model, via an explicit forward process and ELBO comparison, and clarifies why the original non-Markovian VAR is N-th order markovian discrete diffusion. There is thus no overlap in the main technical contributions: HMAR focuses on architectural and systems improvements, while we focus on a establishing a connection between diffusion theory with scale-wise VAR that is absent from HMAR.
> >
> > Although our method uses techniques such as CFG, token resampling, and distillation, these are \emph{not} engineering heuristics in our setting.   Their use becomes principled and mathematically justified only because we prove that Markovian VAR is precisely a discrete diffusion model.   This is the core technical contribution and the key difference relative to HMAR.
> >
> > In more concrete terms:
> >
> > **Our paper introduces a new prescriptive  to look scale-wise VAR as a scale-wise discrete diffusion process---that does not appear in HMAR** To the best of our knowledge, we are first work to connect a variant of VAR to a discrete diffusion. We explicitly define a forward corruption chain $q(x_n\mid x_{n+1}) = M(n)\,x_{n+1}$ and find that in case of deterministic transition matrix the ELBO whose KL terms reduce exactly to the SDD training loss. HMAR does not formulate any diffusion process, any forward kernel, or any variational objective.
> >
> >
> >
> > **Our “optimizations’’ become grounded in diffusion theory under this formulation, not heuristics.**
> >
> > **Classifier free guidance (CFG)** In SDD, each scale transition $p_{\theta}(x_{n-1}\mid x_n)$ is a reverse kernel of a discrete diffusion process. Classifier-free guidance therefore corresponds exactly to scaling the reverse-step likelihood ratio, $\log p_{\theta}(x_{n-1}\mid x_n, y)\;-\;\log p_{\theta}(x_{n-1}\mid x_n)$ which reproduces the monotonic FID/IS behavior established in diffusion theory. Figure~8 matches these predictions precisely: FID decreases from cfg1 $\rightarrow$ 5 and IS peaks at cfg5, followed by saturation. In contrast, VAR are non-Markovian and lack a well-defined reverse-transition likelihood, so CFG in those models is purely heuristic (consistent with VAR collapsing under strong guidance).
> >
> > **Token resampling.** In SDD, resampling is not an ad-hoc trick but is mathematically identical to iterative masked denoising in discrete diffusion, i.e., repeatedly applying the same Markov reverse kernel. This provides a principled explanation for the robustness and stability improvements observed in SDD+MR/SR in Fig. 6 and Fig. 7.
> >
> > **Distillation.** In SDD, skipping scales corresponds exactly to diffusion distillation (consistency models), reducing the number of reverse transitions. This follows directly from the Markovian factorization and mirrors established diffusion-distillation theory.
> >
> > This theoretical clarification is novel and absent in all prior work and this new prescriptive also allow the community to think apart from traditional discrete diffusion to scale-wise discrete diffusion which is theoretically grounded and get better fidelity as shown in the Table above.

---

### Official Review · Reviewer_YJyd · 2025-11-01

**Soundness:** 2
**Presentation:** 2
**Contribution:** 2
**Rating:** 2
**Confidence:** 4

**Summary:**

The paper re-examines Visual Autoregressive Generation (VAR), a popular framework for image generation in the lens of diffusion. By design, VAR encodes different scales of the same image into a sequence of discrete latents, then uses an autoregressive model to model the sequence. The authors argue that a modified "Markovian" version of VAR, which predicts the next image scale based only on the immediate previous scale, is mathematically equivalent to a special case of discrete diffusion model where the transition is deterministic. The paper, in addition, shows that inference techniques applied in diffusion are applicable to "Markovian" VAR and obtains superior performance than the original VAR.

**Strengths:**

The paper elucidates a connection between autoregressive scale generation and discrete diffusion.

**Weaknesses:**

The arguments presented for VAR being discrete diffusion are more conceptual than mathematical. In addition, the formulation can be better explained and mathematical notation should be aligned to that of diffusion.

It's unclear to me why block-wise attention VAR is equivalent to discrete diffusion while VAR is not.

The paper claims that the superior performance of the Markovian attention in VAR is more fundamental (turning VAR to a diffusion model) than a mere regularization effect. However, I find it not convincing enough. (see questions)

**Questions:**

1. Could the authors clarify whether $I_0$ is the latent of original image at the highest scale or the downsampled image at the lowest scale? From the notation, it seems to be the lowest scale as it defined $I_n \in R^{n \times n}$, but then it also says $M(n) \in R^{n^2 \times N^2}$ is a downsampling operation.

2. The authors describe the transition $M(n)$ as "a non-linear deterministic downsampling operation". Is there a more concrete form of $M(n)$ or it only needs to satisfy $R_n=I_n − (I_{n−1})_{\uparrow(n)}$which is dependent on the upsampling network learned? Authors suggest that "the exact transformation is provided in the supplementary material", which I failed to find in the appendix.

3. Could the author provide a clear definition of "Markovian" VAR in the paper?

4. Could the author explain why Markovian attention VAR is equivalent to discrete diffusion while VAR is not? Using a Markovian attention pattern doesn't make the network strictly Markovian $p(R_{n}|R_{n-1})$, since the tokens from previous scale aggregates information from all previous scales. Therefore, the Markovian attention VAR still models $p(R_{n}|R_{n-1},\dots, R_0)$ which is exactly the same as VAR.

5. Could you explain why Markovian VAR results in a higher ELBO than autoregressive formulation? (line 253-255)

6. line 314, is it a typo $beta_2=0.05$?

7. How do the experiments demonstrate the better performance from Markovian attention is not a regularization effect? In the scaling experiment, the gap shrinks as the model gets bigger. The author also admits that the all ablations are on reduced datasets which is a setting strongly favors a regularized model.

---

> ### Author Response · Authors · 2025-11-22
>
> ### Q1: **Regarding notation of $I_0$ .**
>
> We thank the reviewer for pointing this out. In our notation, $I_{0}$ denotes the original full-resolution image (highest scale). All subsequent $I_{n}$ are obtained by applying the deterministic downsampling operator $M(n)$ to $I_{0}$. In our case $N$ varies from 1 to N and $I_0 = I_N$. We will clarify these notations in the main paper.
>
> ### Q2: **Clarification on a non-linear deterministic downsampling operation $M(n)$**
>
> Deriving the closed-form formulation of $M(n)$ is a tedious process. $M(n)$ must satisfy the relation $R_n = I_n - (I_{n-1})_{\uparrow (n)}$, and the residuals must add up such that, it allows the original image to be reconstructed through forward pass with a learned network. In our case, the residual corresponds to the VQ-VAE intermediate output.
> We apologize for the confusion regarding the sentence  ``the exact transformation is provided in the supplementary material.'' Our intention was to refer to the inference-time transformation(eq 11), not to an explicit analytical form of $M(n)$. The supplementary material describes inference algorithm of VAR, but does not attempt to provide a closed-form expression for $M(n)$ itself.
>
> ### Q3: **Definition of Markovian VAR**
>
> We call a model Markovian  when the generative distribution over scales
> factorizes as
> $$
> p_{\theta}(x_1,\ldots,x_N)
> = p(x_1)\prod_{n=2}^{N} p_{\theta}(x_n \mid x_{n-1}),
> $$ i.e., the model forms a first-order Markov chain in the scale index $n$. Assuming information leakage through layer of the transformer is minimal, architecturally, we can enforce this by replacing the original block-wise causal  mask with a Markovian attention mask, in which tokens at scale $n$ are
> only allowed to attend to its immediate previous scale $n-1$. Here, $x_1$ is set as stationary state, SOS token. and $x_N$ is the as the Original image.

---

> > ### Author Response · Authors · 2025-11-22
> >
> > ### W2 and Q4(a): **Equivalence of Markovian VAR and VAR to discrete diffusion**
> > We thank the reviewer for this comment. In fact, VAR can also be formulated as the  N-th order Markov discrete diffusion model. Below we formally define both the first order Markov and N-th order Markov discrete diffusion
> > $$
> > p_{\theta}(x_1,\ldots,x_N)
> > = p(x_1)\prod_{n=2}^{N} p_{\theta}(x_n \mid x_{n-1}),
> > $$ which corresponds to a first-order Markov chain. If we instead consider an  N-th order Markovian formulation (with  N=10 in our case), the model becomes
> > $$
> > p_{\theta}(x_1,\ldots,x_N)
> > = p(x_1)\prod_{n=2}^{N} p_{\theta}(x_n \mid x_1, \cdots , x_{n-1} ),
> > $$
> > where $x_1$ is an SOS token, $x_N$ is the original image and $p_\theta$ is the learnable model.
> >
> > **Loss formulation for Markovian VAR.**
> > Given the forward Markov chain and $M(n)$ as non-linear deterministic transition matrix the standard ELBO decomposes to cross entropy loss
> >
> > $$
> > \mathcal{L}=-\mathbb{E}_{q(x_N)}
> > \left[
> > \sum_{n=1}^{N-1}
> > \log p_{\theta}(x_{n+1} \mid x_n)
> > \right].
> > $$
> >
> > **Loss formulation for general VAR (N-th order Markov discrete diffusion).**
> > For the N-th order forward process the ELBO takes exactly the same discrete-diffusion form, but with higher-order conditionals. Similarly to Markov formulation in  standard ELBO decomposes for N-th order markov  to
> >
> > $$
> > \mathcal{L}
> > =-\mathbb{E}_{q(x_N)}
> > \left[
> > \sum_{n=1}^{N-1}
> > \log p_{\theta}(x_{n+1} \mid x_n \cdots x_1)
> > \right]
> > $$

---

> > > ### Author Response · Authors · 2025-11-22
> > >
> > > ### Q4(b): **Using Markovian attention pattern doesn't make the network strictly markovian**
> > >
> > > We agree with the reviewer that using a Markovian attention mask does not remove information from
> > > earlier scales, since each representation $R_n$ is computed from all previous scales $R_{1:n-1}$.
> > > However, this does not contradict Markovianity.
> > >
> > > In discrete diffusion models (e.g., D3PM), the noisy variable $x_t$ is also a function
> > > of the entire history,
> > > $$
> > > x_t = x_{t-1} Q_t = Q_t Q_{t-1} \cdots Q_0 x_0,
> > > $$
> > > and therefore encodes information from all earlier latent states. Nevertheless, diffusion models are first-order
> > > Markov because their probabilistic factorization satisfies
> > > $$
> > > p_\theta(x_{t-1} \mid x_{0:t}) = p_\theta(x_{t-1} \mid x_t),
> > > $$
> > > meaning that the current state $x_t$ acts as a sufficient statistic of the past.
> > >
> > > Our Markovian VAR (SDD) follows exactly the same principle. Although the multiscale representation
> > > $$
> > > R_n = M(n) M(n+1) \cdots M(N) x_N
> > > $$
> > > contains information from all coarser levels, the generative model is explicitly defined as a
> > > first-order Markov chain:
> > > $$
> > > p_{\theta}(x_1,\ldots,x_N)
> > > = p(x_1)\prod_{n=2}^{N} p_{\theta}(x_n \mid x_{n-1}),
> > > $$
> > >
> > > This structure is fully analogous to the reverse Markov property in discrete diffusion (D3PM):
> > > the representation $R_n$ may contain information from all earlier states, but the
> > > \emph{probabilistic model} is first-order Markov. Therefore, Markovian attention VAR is
> > > equivalent to discrete diffusion even though its internal representations aggregate information
> > > from earlier scales.
> > >
> > > Alternatively, here we present another way to  approach at this question. Consider the scales at levels $n$ and $n+2$.  If information leakage were to happen under the Markovian masking, it must first pass from $n$ to $n+1$, and then from $n+1$ to $n+2$.   Thus, the total transfer from $n$ to $n+2$ is upper-bounded by $\frac{N^2}{(N+2)^2 + N^2}$, this is through the tokens that interact in the attention layer. Moreover, the residuals themselves may be approximately orthogonal, further reducing this leakage in practice.  This proportion is significantly smaller than in standard VAR, where direct information transfer occurs across all previous scales.   Therefore, our formulation is much closer to a Markovian process compared to VAR which is very similar to diffusion model.

---

> > > > ### Author Response · Authors · 2025-11-22
> > > >
> > > > ### Q5: **Explanantion of why Markovian VAR results in a higher ELBO than autoregressive formulation**
> > > >
> > > > In the autoregressive VAR (N-th order Markov discrete diffusion), the model learns the higher-order conditional  $p_\theta(R_{n-1}\mid R_n,\ldots,R_0)$, where the inputs  $R_{0:n-1}$ are model-generated. If there occurs a errors at coarser scales then it will get propagate forward, since each prediction is conditioned on previously generated scales.  As a result, the model faces a uncertain prediction and the KL terms become larger, leading to a looser ELBO. In contrast, the Markovian VAR only predicts $R_{n-1}$ from the current scale $R_n$, avoiding error accumulation and therefore achieving a higher ELBO.
> > > >
> > > > ### Q6: **Typo regarding the AdamW**
> > > > We apologize for the typo, we use AdamW optimizer with $\beta_{1}=0.9$, $\beta_{2}=0.95$ and weight decay $=0.05$.

---

> > > > > ### Author Response · Authors · 2025-11-22
> > > > >
> > > > > Thank you for your valuable time and insightful comments.  We have tried to address your concerns in our rebuttal text:
> > > > >
> > > > > ###Q7: **Experimental results on ImageNet256x256 and comparison with existing state-of-art discrete diffusion and image diffusion models**
> > > > >
> > > > > We agree with the reviewer that evaluating at scale is important. To address this concern, we performed an additional full ImageNet-256×256 experiment using the same experimental setup. Under our resource constraints, this required **21 days of continuous training on 4×A6000 GPUs.**
> > > > >
> > > > > | Method       |  FID  |   IS  |
> > > > > | :------------| ----: | ----: |
> > > > > | DiT-L/2      | 5.02  | 167.2 |
> > > > > | LDM-4-G      | 3.60  | 247.7 |
> > > > > | MaskGIT      | 6.18  | 182.1 |
> > > > > | RQTran       | 7.55  | 134.0 |
> > > > > | Dart-AR      | 3.98  | 256.8 |
> > > > > | ReMDM*       | 7.40  | 145.2 |
> > > > > | MD4*         | 6.51  | 246.3 |
> > > > > | VAR          | 3.36  | 274.4 |
> > > > > | Ours         | 3.06  | 286.6 |
> > > > >
> > > > > For fair comparisons, we compare with the most recent discrete-diffusion models (*ReMDM and *MD4 [1]) and AR (Dart-AR) models for image generation on ImageNet dataset.
> > > > >
> > > > > The results on ImageNet portray the scalability of SRDD, even with the availability of very less compute showing that the discrete-diffusion reinterpretation is beneficial beyond the smaller datasets in the main paper and without relying on any training-time heuristics.
> > > > >
> > > > > A distinguishing feature from ReMDM and MD4, which are state-of-the-art discrete diffusion models is that these models are built directly in the traditional D3PM framework. In contrast, our method starts from a scale-wise VAR architecture via the Markovian masking and a discrete diffusion model reinterpretation. Our approach also follows structured, deterministic transitions across scales. The full scale ImageNet results show that this new perspective does not only unify VAR and discrete diffusion theoretically, but also delivers significant performance gains over existing discrete-diffusion approaches. These results, together with our theoretical reinterpretation, suggest that the traditional discrete-diffusion framework itself can be rethought in a scale-wise manner, potentially enabling new classes of efficient multi-scale discrete diffusion models.
> > > > >
> > > > > [1]. Informed Correctors for Discrete Diffusion Models

---

### Official Review · Reviewer_4Y57 · 2025-11-03

**Soundness:** 3
**Presentation:** 3
**Contribution:** 3
**Rating:** 6
**Confidence:** 3

**Summary:**

The paper shows that Visual Autoregressive Generation (VAR) becomes mathematically equivalent to a discrete diffusion process if you replace its block-causal mask with a Markovian attention mask (each scale conditions only on the previous one). Framed as Scalable Discrete Diffusion (SDD)—and with confidence-aware masked resampling and classifier-free guidance (SRDD)—the reinterpretation lets the authors import standard diffusion tricks (iterative refinement, guidance, scale pruning) directly into VAR. Empirically, SRDD improves sample quality and efficiency across 256×256 benchmarks.

**Strengths:**

1. The paper gives a clean connection from scale-wise AR to discrete diffusion.
2. Based on this connection, several training techniques proposed in discrete diffusion can be introduced to improve the performance of the Markovian version of VAR.
3. SRDD improves over VAR across datasets and is better in/out-painting and super-resolution without task-specific training.

**Weaknesses:**

1. Experiments are limited to small datasets; comparisons against current strong scale-wise AR or modern discrete-diffusion generative models at similar compute are absent.
2. No precise theorem that states conditions under which the Markovian VAR is exactly a discrete diffusion.
3. There are several typos and formatting errors in the paper. Please polish the paper writing.

**Questions:**

None

---

> ### Author Response · Authors · 2025-11-22
>
> Thank you for your valuable time and insightful comments.  We have tried to address your concerns in our rebuttal text:
>
> ### W1: **Experimental results on ImageNet256x256 and comparison with existing state-of-art discrete diffusion and image diffusion models**
>
> We agree with the reviewer that evaluating at scale is important. To address this concern, we performed an additional full ImageNet-256×256 experiment using the same experimental setup. Under our resource constraints, this required **21 days of continuous training on 4×A6000 GPUs.**
>
> | Method       |  FID  |   IS  |
> | :------------| ----: | ----: |
> | DiT-L/2      | 5.02  | 167.2 |
> | LDM-4-G      | 3.60  | 247.7 |
> | MaskGIT      | 6.18  | 182.1 |
> | RQTran       | 7.55  | 134.0 |
> | Dart-AR      | 3.98  | 256.8 |
> | ReMDM*       | 7.40  | 145.2 |
> | MD4*         | 6.51  | 246.3 |
> | VAR          | 3.36  | 274.4 |
> | Ours         | 3.06  | 286.6 |
>
> For fair comparisons, we compare with the most recent discrete-diffusion models (*ReMDM and *MD4 [1]) and AR (Dart-AR) models for image generation on ImageNet dataset.
>
> The results on ImageNet portray the scalability of SRDD, even with the availability of very less compute showing that the discrete-diffusion reinterpretation is beneficial beyond the smaller datasets in the main paper and without relying on any training-time heuristics.
>
> A distinguishing feature from ReMDM and MD4, which are state-of-the-art discrete diffusion models is that these models are built directly in the traditional D3PM framework. In contrast, our method starts from a scale-wise VAR architecture via the Markovian masking and a discrete diffusion model reinterpretation. Our approach also follows structured, deterministic transitions across scales. The full scale ImageNet results show that this new perspective does not only unify VAR and discrete diffusion theoretically, but also delivers significant performance gains over existing discrete-diffusion approaches. These results, together with our theoretical reinterpretation, suggest that the traditional discrete-diffusion framework itself can be rethought in a scale-wise manner, potentially enabling new classes of efficient multi-scale discrete diffusion models.
>
> [1]. Informed Correctors for Discrete Diffusion Models

---

> ### Author Response · Authors · 2025-11-22
>
> ### W2: **Theoretical Equivalence of Markovian VAR and Discrete Diffusion**
>
> **Definition (Markovian VAR).**
> Let $(x_0,\dots,x_N)$ denote the sequence of discrete latent images (or token grids)
> from the coarsest scale $1$ to the finest scale $N$, where $x_0$ and $x_N$ denote the original image.
> For each $n=1,\dots,N$, let $M(n)$ be a non-linear deterministic downsampling transition
> mapping $x_{n+1}$ to the coarser representation $x_n = M(n)\, x_{n+1}$.
>
> A *Markovian VAR* model specifies:
> - a prior $p(x_N)$ over the finest scale;
> - a final stationary state $x_1$, which is $\langle SOS \rangle$ in our case;
> - for each $n=1,\dots,N$, a conditional distribution  $$ p_\theta(x_{n+1}\mid x_n),$$ parameterized by a transformer that is allowed to attend
>   *within* scale $n$ and only to the immediately coarser scale $n$
>   (Markovian attention across scales).
>
> Generation proceeds by sampling,
> $$
>  x_1 \sim p(x_1), \qquad
>  x_{n+1} \sim p_\theta(x_{n+1}\mid x_n)\quad \text{for } n=1,\dots,N.
> $$
>
> ---
>
> **Definition (Discrete diffusion model).**
> Let $(x_0,\dots,x_N)$ be a sequence of discrete states and let $Q_n$ be
> an absorption transition matrix. Here $x_0$ is the original image.Define a forward Markov chain
> $q(x_{1:N}\mid x_0)$ by
> $$
>    q(x_n \mid x_{n-1}) = x_{n-1} Q_n, \qquad n=1,\dots,N,
> $$
> i.e., $x_n$ is obtained by transitioning $x_{n-1}$ through $Q_n$.
>
> Given a prior $p(x_N)$ and reverse kernels $p_\theta(x_{n-1}\mid x_n)$,
> the associated *discrete diffusion model* is the joint distribution
> $$
>    p_\theta(x_0,\dots,x_N)
>    = p(x_N)\prod_{n=1}^N p_\theta(x_{n-1}\mid x_n).
> $$
>
> ---
>
> **Proposition (Markovian VAR is a discrete diffusion model).**
>
> Consider a Markovian VAR model as in Definition 1 and the
> discrete diffusion model in Definition 2. Assume:
>
> 1. **Deterministic forward chain.**
>    For all $n=1,\dots,N$,
>    $$
>      q(x_n\mid x_{n+1}) = M(n)\, x_{n+1},
>    $$
>    i.e., the forward process is the fixed non-linear downsampling transition
>    induced by $M(n)$.
>
> 2. **Shared reverse kernels.**
>    The reverse kernels $p_\theta(x_{n-1}\mid x_n)$ in the diffusion model
>    coincide with the Markovian VAR conditionals $p_\theta(x_{n+1}\mid x_n)$.
>
>
>
> 3. **Markovian VAR training loss.**
>    The Markovian VAR is trained with the token-wise cross-entropy loss
>    $$
>        L_{\\mathrm{o}}(\\theta)  = - \sum_{n=1}^N E_{q(x_N)} \,E_{q(x_n\mid x_N)} \log p_\theta(x_{n+1}\mid x_n),
>
>    $$
>    where $q(x_n\mid x_N)$ is obtained by running the deterministic forward chain
>    from $x_N$ down to scale $n$.
>
>
>
> ---
>
> Then:
>
> 1. The joint distribution of the Markovian VAR model is exactly a discrete
>    diffusion model with forward kernel $q$ and reverse kernel $p_\theta$:
>    $$
>       p_\theta(x_0,\dots,x_N)
>       = p(x_N)\prod_{n=1}^N p_\theta(x_{n-1}\mid x_n).
>    $$
>
> 2. The discrete diffusion variational bound
>
>    $$
>    \begin{aligned}
>    \mathcal{L}_{\mathrm{v}}
>    &= \mathbb{E}_{q(x_0)} \Bigg[
>        D_{\mathrm{KL}}\!\bigl(q(x_T \mid x_0)\,\|\,p(x_T)\bigr)
>        + \sum_{t=2}^{T}
>          \mathbb{E}_{q(x_t \mid x_0)}
>          D_{\mathrm{KL}}\!\bigl(q(x_{t-1}\mid x_t, x_0)\,\|\,p_\theta(x_{t-1}\mid x_t)\bigr)
>    \\[2pt]
>    &\qquad\qquad\qquad\qquad
>        -\,\mathbb{E}_{q(x_1 \mid x_0)}\!\left[\log p_\theta(x_0 \mid x_1)\right]
>    \Bigg]
>    \end{aligned}
>    $$
>
>    satisfies
>
>    $$
>         \mathcal{L}_{\mathrm{v}}(\theta)
>         = \mathcal{L}_{\mathrm{o}}(\theta) + C,
>    $$
>    where $C$ is a constant independent of $\theta$.
>
> ---
>
> In particular, minimizing the Markovian VAR training loss is exactly equivalent
> to maximizing the discrete diffusion ELBO.
> Thus, Markovian VAR is a discrete diffusion model in the D3PM sense with
> deterministic forward transitions.
>
> ### W3: **Typo and formatting errors**
> Thank you for bring this to our attention. We will carefully proof-read the entire manuscript and fix all typos and formatting errors.

---

> ### Author Response · Authors · 2025-11-28
>
> As OpenReview’s formatting constraints limit the presentation of certain mathematical expressions, we have also included the answer of the question “Theoretical Equivalence of Markovian VAR and Discrete Diffusion” in Section D of the supplementary material.

---

### Note · Program_Chairs · 2026-01-17
**Submission Desk Rejected by Program Chairs**

The following references in this submission do not refer to real documents and/or have major errors in bibliographic information:

 Runtian Luo, Lingkai Kong, Shaojie Wei, Wenhao Chen, Shixiang Shane Gu, and Yue Zhang. Gibbsddpm: Generating discrete data with a denoising diffusion model. In International Conference on Learning Representations (ICLR), 2023a. URL https://arxiv.org/abs/2202.00817.